# A Multiagent System Prototype of a Tacit Knowledge Management Model to Reduce Labor Incident Resolution Times

**Lilyam Paolino [1,2], David Lizcano [3,*] , Genoveva López [2] and Jaime Lloret [4]**

1   School of Engineering, University ORT Uruguay, Montevideo 11100, Uruguay; lilyam.paolino@gmail.com
2   School of Computer Science, Universidad Politécnica de Madrid, Campus Montegancedo,
    28660 Madrid, Spain; glopez@fi.upm.es
3   School of Computer Science, Madrid Open University, UDIMA, 28400 Madrid, Spain
4   Instituto de Investigación para la Gestión Integrada de Zonas Costeras (IGIC), Universitat Politecnica
    de Valencia, 46022 Valencia, Spain; jlloret@dcom.upv.es
*   Correspondence: david.lizcano@udima.es

**Abstract:** The transformation of the tacit knowledge of a company's human resources into permanent organizational capital in spite of possible staff turnover is of business interest. This research focuses on the management of tacit knowledge to resolve labor incidents and reduce resolution times. We present the GESTAC model, a name derived from the first syllables of the Spanish words "gestión" (management) and "tácito" (tacit), which identifies, locates and rates people in the business domain capable of resolving a labor incident logged by a user employed by the company. In order to achieve its objective, the GESTAC model follows the tacit knowledge management paradigm, according to which tacit knowledge that could eventually resolve the logged incidents is identified, captured and stored in a permanent database, and then evaluated and disseminated to the people who have need of the knowledge. This could lead to the knowledge source being automatically rerated, and the entire process restarted. The aim is to contribute to the state of the art, showing that by applying tacit knowledge management to a specific domain the GESTAC model is able to reduce incident resolution times with respect to traditional systems. The model was prototyped (GESTAC_APP) using the multiagent systems paradigm.

**Keywords:** incidents; response times; tacit knowledge; knowledge sources; multi-agent systems

## 1. Introduction

The tacit knowledge of the human resources of companies is a precious asset for them, which often cannot be exploited in an adequate time and manner because it is implicit, not formalized and not communicated. This article proposes a model to transform this tacit knowledge into permanent organizational capital, properly maintained and available, and validates this model in an area where this knowledge is especially necessary: the resolution of labor incidents. The research will focus on knowledge management in the business world, arising from "know how" in business environments, applied as a solution to labor incidents logged by users. It focuses on unstructured knowledge stored subconsciously and applied automatically (almost unconsciously), that is, tacit knowledge, based on interpersonal relationships and internal communication processes. It is also concerned with how the multi-agent system paradigm can help with its management.

This research addresses the problem of what advantages the application of tacit knowledge management has in the workplace. To do this, we have to identify the knowledge and the knowledge proprietors, and store and socialize the knowledge in the business domain to assure that it is not

lost to the company if the link between the knowledge and its proprietor is severed. We take up the idea that tacit knowledge management makes it possible to turn unreflective into reflective practice, facilitating the emergence of heuristic knowledge, bringing to the surface the norms that govern collective work practices [1]. Knowledge is understood as a dynamic intangible resource, which must be permanently updated. For this purpose, a tacit knowledge management model was designed, called GESTAC, a name that originally resulted from joining the first two syllables of two Spanish words, "gestión" (management) and "tácito" (tacit), which succinctly define the objective of the model. GESTAC aims to convert tacit into explicit knowledge. It should evaluate knowledge and knowledge sources permanently. A prototype was implemented to validate the model in a specific business situation (GESTAC_APP).

GESTAC accounts for particular factors that emerged from the exploration of knowledge management (KM) expert opinions regarding the selection of knowledge sources, the rating of the explicit knowledge and knowledge used to resolve labor incidents. Expert opinions were gathered by applying the Delphi method [2]. The key functions are implemented by means of the multi-agent systems paradigm, considering that software agents are programs that can choose the action to be performed and learn from perceptions of a static or dynamic, discrete or continuous, total or partially observable environment [3].

This research regards tacit knowledge management as a tool not merely to support business strategies, but also to contribute to innovation through the continuous acquisition of new and better knowledge [4]. It was hypothesized that the GESTAC model could help improve response times to labor incidents, compared with systems that do not use knowledge management.

The paper is structured in seven sections. Section 2 describes the theoretical groundwork underlying the research and some of the knowledge management models used in business domains. It highlights the differences between previous models and the proposed GESTAC model. Section 3 describes materials, methods, the GESTAC model, its objective, the hypothesis to be tested, its design features, components and how they interact, as well as the processes and associated calculations for evaluating and continually reevaluating knowledge and knowledge sources. Section 4 details the methodology used throughout the incident handling processes, and the model design, layers, objectives and activities. Section 5 explains GESTAC model prototype, its objective, its architecture, the reason for using the multiagent systems paradigm, the software agents used, the logic of some of the algorithms and their main encodings. It also provides specific examples of inter-agent communication, and addresses agent interaction, domain layer agents, the data access agent, and the class diagram. Section 6 specifies the aims of the experimentation, its theoretical background, its design, the incidents used in the experiment and the analysis of the results according to different statistical tests to accept or reject the proposed hypothesis. Section 7 analyzes results, whether or not the implemented prototype meets the original objective, and whether the project generated advantages in the business world. Section 8 describes possible future lines of research concerning the GESTAC model.

## 2. Background

This section discusses the opinions of some researchers on the importance of different types of knowledge in business domains and describes what differentiates GESTAC from some related knowledge management models.

### 2.1. Definition of Tacit Knowledge

The present research considers that tacit knowledge is that implicit, not formalized and not communicated wisdom that knowledge workers treasure thanks to their day-to-day experience [1]. Thanks to this knowledge, they are able to carry out tasks quickly and efficiently, without having to invest too much time in thinking about what steps to take, or how to carry them out, since they do so almost systematically.

## 2.2. Importance of Knowledge in Business Undertakings

Nonaka [5] classified knowledge as explicit (it can be structured, recorded, stored and distributed) and tacit (it is part of know-how gathered through individual experiences and personal learning, is hard to structure, store in repositories and distribute). He claimed that, shared knowledge cannot be easily leveraged by the organization as a whole unless it is made explicit [1]. The Financial Accounting Standards Board (FASB) international accounting rules published in December 1984 [6] considered tangible resources—capital, labor, land, etc.—, as factors in the effective management of assets. According to a study by the Brookings Institute cited by Kaplan [7], intangible resources, like knowledge, skills and capabilities, accounted for 38% of the market value of industrial organizations in 1982, rising to 62% in 1992. Knowledge sourced from individual minds is a mixture of experiences, values and know-how [8]. It constitutes a major business resource by generating new experience and knowledge by means of a life cycle that passes through the stages of generation, access, storage and transfer. According to Thompson, Peteraf, Gamble and Strickland [9], it is intangible resources that generate a company's competitive edge.

Knowledge specification in Nonaka's model [10] is a process that can involve the processing of each individual's particular ideas expressed as images, metaphors or analogies. It includes tacit knowledge transfer from others gathered by means of face-to-face interpersonal dialog processes as a means of sharing ideas and learning by exchanging experiences within work groups.

According to Probst, Raub, and Romhardt [11], knowledge is the whole body of cognitions and skills that individuals use to solve problems. These cognitions and skills reflect their beliefs and are composed of theory, practice, routine rules and instructions for action. Chiavenato [12] claims that the achievement of business goals is linked with knowledge management, understood as the process of planning, organizing, directing and controlling the use of organizational resources. Wiig [13] alleges that the value of knowledge lies in its use. It is influenced by information technology (IT) developments, which facilitate pragmatic knowledge management, and by the multiagent system paradigm, which provides for experimentation with the models. Noh, Lee and Kim [14] demonstrated that knowledge reuse was useful in business domains, using cognitive maps representing individual mental reasoning. The cognitive maps are related to each other by causal effects. Martínez [15] concluded that the ultimate aim of all KM is action as the basis for decision making in any business domain. Tools like content management systems have been developed to help collect, structure, store and distribute explicit knowledge [16]. However, tacit knowledge collection, structuring, storage and distribution still pose a challenge. GESTAC aims to formalize tacit knowledge on solutions to business incidents.

## 2.3. Models Related to GESTAC

From an ontological perspective, Nonaka et al. [5], regard individuals to be the source of organizational knowledge, whereas the company should act as a facilitator to stimulate the capture of the knowledge and its later distribution [10]. Tacit knowledge management transforms individual knowledge into organizational knowledge. One of the questions to be defined as part of the operational strategies is what knowledge the management system should operationalize. Nonaka et al. claim that one important aspect of tacit business knowledge management is to stimulate the motivation of individuals to generate new knowledge. Therefore, they recommend promoting individual autonomy within the circumstances of the business, stimulating the individual–context interaction (internal and external to the company), taking advantage of the emergence of creative chaos to find creative solutions to problems, and promoting redundant problem-solving proposals to assure that the best knowledge emerges in the confrontation. This model developed a theory that set the pace in the knowledge management area. However, it fell short of implementing the knowledge transfer processes, which it dealt with too superficially.

Wiig's model [13] describes different levels of knowledge internalization by its sources, ranging from novice to expert. He does not make a big enough distinction between ontologies (study of being) and epistemologies (study of scientific knowledge).

Al-Mutawah's model [17] focuses on the management of tacit knowledge for product manufacturing processes, calculating knowledge by means of a template formula according to which the same activity can share environments integrated based on knowledge management by different agents. This model overcomes the knowledge management bottleneck by locating the data to be processed. However, it was applied exclusively in the production domain. Okike, Fernandes and Xiong [18] weighted the previous knowledge transfer and reuse processes experience, whereas GESTAC considers that this is not the only item to be considered. Gal's model [19] shows that tacit knowledge drives actions, although there is often no rigorous analysis enabling its application by other actors. Although it deals with the interaction between agents, this model does not take into account socializing elements in inter-agent relations. Seidler, de Alwis and Gemunden [20] consider that tacit knowledge is the key to innovation processes, but do not model its acquisition or externalization. Brigui-Chtioui and Saad's model [21] focuses on the management of what is regarded as essential or crucial knowledge by a mediator agent that regulates the conflicts occurring in business decision making and sets out the integration of multi-agent systems as suitable knowledge management paradigms. However, they only consider explicit knowledge. Karunananda, Ratnayke and Mendis's model [22] uses Visual Basic and fuzzy logic in expert systems to specify tacit knowledge. GESTAC considers that Visual Basic is probably not the best tool to achieve these goals in the distributed, dynamic, flexible and frequently imprecise environments in which they are applied. Grant's model [23] focuses on analyzing the knowledge created by individuals employed by the organization, but does not examine the existence of organizational knowledge at length. Hedlund and Nonaka's model complies with the stages of the knowledge management life cycle, providing useful techniques for storage, such as the knowledge encyclopedia which operates like a repository [24]. This model details the knowledge management phases, but it does not specify how knowledge is created.

GESTAC is a different proposal to the above models, because it brings together several areas of business interest: tacit knowledge, labor incident resolution, knowledge management and the multiagent systems paradigm, including both the model design and prototype implementation (GESTAC_APP). This prototype was used to validate the model in real environments. Knowledge management is not easy to apply to business practices, because it takes time, which enters into conflict with the pressure of routine job performance [25]. In an attempt to integrate knowledge management into the business field, GESTAC applies it to incident resolution by trying to convince users of its benefit in shortening resolution times. This was the hypothesis that the experimentation of the GESTAC model was to test. GESTAC is able to conserve the tacit knowledge of company employees, provide access to the best solution for each incident, stimulate the individual growth of each worker, and continually and automatically evaluate possible knowledge sources and the knowledge provided by the multiagent system paradigm without the need for human intervention.

## 3. Gestac Model: Material and Methods

### 3.1. Theoretical Basis of the Model

GESTAC is based on systematic tacit knowledge management. Tacit knowledge is personal knowledge. Tacit knowledge is unlimited because it expands the more it is used. This highlights that there is a directly proportional relationship between knowledge and learning: the greater the flow of knowledge, the greater the learning capacity [26].

Tacit knowledge management includes the ability to create, identify, capture and transfer knowledge so that it can be accessed by the rest of the company. Knowledge is created through interaction with other people and with the environment [10]. Tacit knowledge is linked to the way people behave [27,28].

Later research has found tacit knowledge to be related to people's hands, minds, and specifically takes into account the importance attached to the beliefs, emotions, and feelings of the knowledge sources in its management. For Broncano [29], there are two approaches to knowledge:

so-called true knowledge (which is the object of philosophical research) and useful knowledge (which concerns companies). According to Broncano, certain conditions are required for tacit knowledge management to be successful: human interaction, autonomy among those involved, physical possibilities of face-to-face interaction understanding and acceptance of business guidelines, interaction with other people and the environment, the form of knowledge creation.

Aguilera and Luke [30] propose a tacit organizational knowledge management model that includes the following phases: tacit knowledge sharing, concept creation and justification, architecture creation and finally distribution. Knowledge must be valued and classified as relevant to the company for production. The knowledge generated is integrated into a specific business group, field or domain.

The GESTAC model captures the tacit knowledge of all the members of each business area by stimulating the relationship between the people who hold and the people who need the knowledge. GESTAC relates the search for labor incident solutions with the knowledge that company members possibly hold. GESTAC takes into account only the knowledge defined as essential for the company framed within the strategic tasks of each area. It considers technology not as an end but as a means for recording and distributing the captured knowledge so that the model can be extended to any other type of company. The GESTAC application assumes that knowledge-related processes are dynamic [31].

This model is innovative in that it aims to stimulate the link between the people who need and the people who hold the knowledge by relating labor incidents logged in their respective areas and captured tacit knowledge. Additionally, it aims to always dynamically search for the best knowledge, adapting to possible changes in the environment, also making the knowledge source responsible, through evaluation, for the level of knowledge confidence and veracity. Another original point is that it aims to contribute to organizational knowledge through member tacit knowledge management. The model prototype, called GESTAC_APP, aims to evaluate whether the model was successful in this regard with respect to the specified use case. Although the model was applied to a specific case, the model was designed and implemented to be applicable to any other company type operating in any other sector by merely changing the input data.

### 3.2. Model Description

Figure 1 shows the logical processes of the model from incident registration to resolution. Users log the incidents. The incidents then have to be identified or, if they do not exist, entered in the knowledge base (KB). If the KB contains the incident but no recorded resolution, a knowledge discovery process is launched to identify the best possible source for capturing, weighting and recording tacit knowledge, which is made accessible for use and distribution within the business domain and rated by each user.

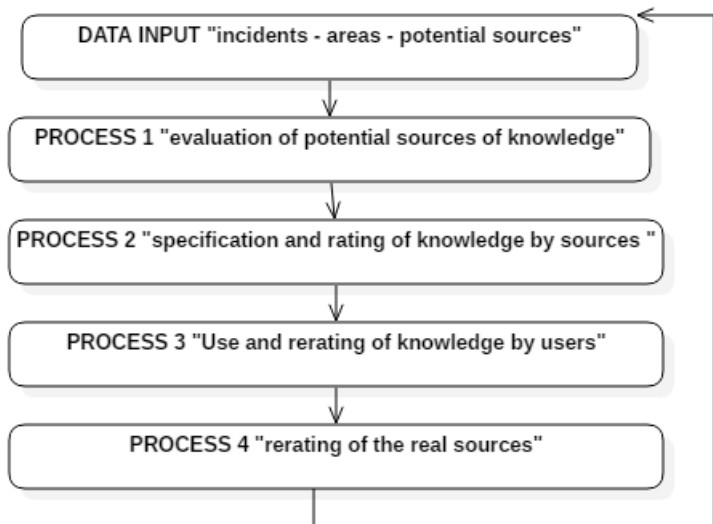

**Figure 1.** The logic of the GESTAC model.

The potential knowledge sources are evaluated, as shown under Process 1 in Figure 1. This process is based on a trust model, whose objective is to discover the real sources. Once the knowledge has been identified and captured, it is continually rated, recorded, used and rerated.

### 3.3. Model Goals and Hypothesis to Be Tested

Focusing on the tacit knowledge of people connected to business environments, the main aim is to offer dynamic and possibly changing solutions to any workplace incidents logged by users by means of specification, storage, use, continual rating and distribution.

The hypothesis to be tested is whether, by enabling tacit knowledge discovery through the identification of knowledge sources, knowledge capture and continual knowledge rating by autonomous software agents, the GESTAC model can have a positive impact on the resolution times of business incidents logged by users.

### 3.4. Design Features and Model Components

GESTAC was designed modularly and cyclically, whereas each module includes different processes. It is cyclical because, as it always searches for the best knowledge, the weighting processes start automatically when new components appear. The model components are:

a. Area. The area refers to the sector of the business or organization. It does not necessarily match up with an area or department in the organizational chart, but is rather a cluster identifying related agents, a knowledge community working on one and the same knowledge domain.

b. Incidents. According to the ITIL (Information Technology Infrastructure Library) framework [32], an incident is "An unplanned interruption to an IT service or reduction in the quality of an IT service (situation to be resolved)", and severity is the level of difficulty involved in the incident. The variables *subject* and *severity* applied in the conceptual design were taken from our previous research [33]. Each incident is specific to one area.

c. Potential sources. All the experts connected to the area of the logged incident. They are subject to evaluation processes designed to define the best rated source in the area.

d. Real source. The best rated potential source in the area, which is presumed to possibly have the sought-after tacit knowledge, which will be corroborated or rectified in later rerating processes.

e. Tacit knowledge captured from the source. This knowledge is rated by its first user, the real source, and by each of the later users when used in incident resolution. Any differences between the two ratings generate rerating processes of the real source. The source and knowledge rating and rerating criteria are the result of the surveys conducted on knowledge management experts and the results of other research.

Figure 2 shows the components of the model (area, incident, source, knowledge and rating) and the cardinality relations between them. These components are processed in the different phases of the GESTAC model in order to meet the objectives of each of the knowledge management phases.

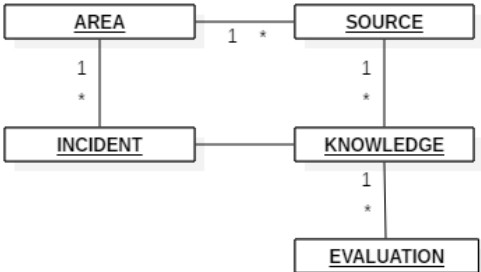

**Figure 2.** Cardinality relations between components, being * a cardinality 1 to N.

### 3.5. Processes Applied to GESTAC Model Components

In this section, we detail the processes illustrated in Figures 1 and 2. The GESTAC process for identifying and capturing the real source evaluates the trustworthiness of each potential source. Choi [34] states that the best personal profiles within a social group can be identified by comparing their behavior and intelligence according to rating tests. Cornu [35] claims that confidence is a hypothesis regarding future behavior. It is a forward-looking attitude insofar as this future depends on the action. For the choice of the source weighting items, GESTAC compares the personal ratings of each source. To do this, it combines some of the criteria based on Soto Barrera's confidence model [36] (hierarchical position in the business and job experience in the area) with items output by surveys taken by personnel managers at different companies conducted during the research (training in the area and perception). This generates a knowledge source confidence model [37]. Confidence in the potential source, Conf(y), is rated as follows:

(1) Source confidence, taken into account by GESTAC, is entered when the person joins the business. It is calculated according to:

    (a) Training in the area (TA(y)): type and length of training that the source received with respect to his or her tasks in the area;

    (b) Job Experience, (WE(y)): time that the source has been performing tasks in the area;

    (c) Position, (P(y)): office or rank held by the source in the organization.

(2) Perception, (PR(y)), is input by the source's immediate superior three months after the potential source joined the area, as this is generally the staff trial period, and then periodically every year, which is the usual performance appraisal period. It is calculated according to the average of five attributes (PRS, PRM, PRI, PRC and PRCG), referring to the potential source's behavioral skills or qualities. The items for evaluating PR(y) emerged as the result of surveys taken and are:

    (a) Common sense (PRS) [38]: extent to which the source applies careful and reasonable judgment to solve problems;

    (b) Methodicality (PRM): ability to efficiently schedule job activities to achieve the expected goals and results [39];

    (c) Interest in the area (PRI): source's liking for or inclination towards a subject, thing or situation [40], in this case with respect to his or her work area;

    (d) Conscientiousness [40] (PRC): ability to understand processes, make decisions and keep calm in the face of problems;

    (e) Teamwork capability (PRCG): commitment, motivation, and communication, negotiating and conflict management skills [41].

The formula applied is:

$$\text{Conf}(y) = (\text{TA}(y) + \text{WE}(y) + \text{P}(y) + \text{PR}(y))/4 \tag{1}$$

$$\text{PR}(y) = (\text{PRS}(y) + (\text{PRM}(y) + (\text{PRI}(y) + \text{PRC}(y) + \text{PRCG}(y))/5 \tag{2}$$

### 3.6. Specification of Tacit Knowledge

Based on the view of specialized researchers, like Shah and Overy [42] who suggested using a list of business member profiles to capture their knowledge, and Stenmark [43] who claim that feedback from question answering can reveal some sort of tacit knowledge, GESTAC captures this knowledge by asking sources how they would resolve the logged incidents or asking them to resolve similar situations that they are very unlikely to have experienced before. Some researchers [44] concluded, based on the analysis of real-life situations, that the interest taken by people in certain tasks or components could be a way of specifying their knowledge. Cheng and Cohen [45] designed systems to extract tacit knowledge

from the behavior of users accessing online information and created knowledge layers, a probabilistic tool that captures tacit knowledge from event logs. GESTAC associates the incident solutions log with the tacit knowledge of the people who provided such solutions. With the aim of improving the Internet-user browser experience, Román [46] uses frameworks and models processes using behavioral patterns to analyze transactions made by users and be able to capture their knowledge.

GESTAC saves the data and outputs of the model processes in a SQL server. The methodology defined in GESTAC involves electronically sending the incidents with no logged solution to the real source in the area. As suggested by Shah and Overy [42], the real source is questioned with regard to possible solutions. The output of this phase is the knowledge specification through the natural language response provided by the source that is logged into a KB [47]. The captured knowledge is stored using Hibernate [48], a tool capable of mapping Java objects to the relational database model and using the database to create, update and query the knowledge. GESTAC integrates Spring Boot as a framework for developing Java applications. Spring Boot speeds up development, minimizing the necessary configuration using an embedded TomCat web server that improves prototype development, use and portability and provides for the possible implementation of a computer system to be run later on a web server.

### 3.7. Evaluation of Tacit Knowledge

Knowledge is evaluated by users. It is first rated by the source that provided the knowledge (C(y)) (3), as this is a special user who should be distinguished from others, and later by users who use the knowledge, C(Xi->y), (4). The final results (5) for knowledge confidence, Cfinal(y), are calculated by adding together the ratings output by applying (3) and (4). The knowledge is weighted according to a combination of two attributes taken from Soto Barrera's model [34] (reuse and resolution time) with a third (simplicity) based on expert opinions stated as survey responses. If any user is unable to solve the incident with the provided knowledge, he or she would rate the knowledge as 0. The items are equally weighted as determined by the results of the surveys taken by experts. The attributes of (3) are defined as follows:

- Reuse (R(y)): number of times the knowledge was used to solve recurrent incidents, as proposed by Smith and Duffy [49].
- Resolution time (TR(y)): time elapsed from when the incident is logged to its resolution according to the ITIL model.
- Simplicity (S(y)): number of activities required to resolve the incident using the knowledge provided according to the results of the surveys taken, referred to by ITIL as incident resolution effort. Figure 3 shows the source and knowledge rerating flow. The system compares the knowledge ratings with each other and saves the best knowledge per incident. The source reratings are determined by the knowledge reratings for sources.

Equation (3) denotes the evaluation of knowledge by source.

$$C(y) = (R(y) + TR(y) + (S(y))/3 \tag{3}$$

Equation (4) denotes the average ratings given by users, and n is the number of users that rated the knowledge. Equation (5) shows the final result after summing both Equations (3) and (4).

$$C(xi->y) = \sum_{i=1}^{n} (((R(xi->y) + TR(xi->y) + S(xi->y)/3)/n) \tag{4}$$

$$Cfinal(y) = (C(y) + C(xi->y))/2 \tag{5}$$

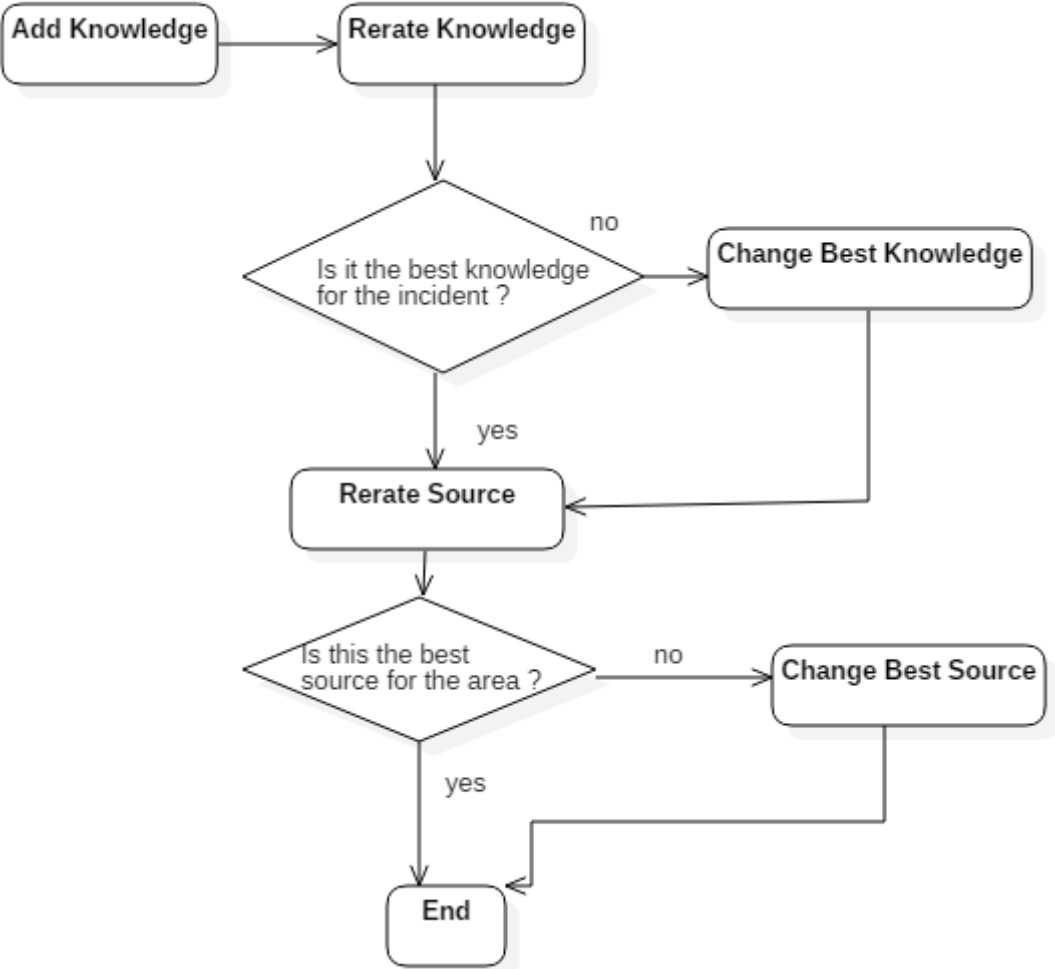

**Figure 3.** Source and knowledge rerating.

### 3.8. Source Rerating as a Result of Knowledge Rating

Figure 3 shows the originality of the model of the relationship between knowledge and its source, designed to encourage sources to provide useful and serious knowledge, since this will influence the source ratings. Source reratings, (RevSources(y)), (6), are determined by the knowledge reratings for sources (Figure 3).

The goal of this model is to account for the influence of knowledge quality on the source's rating. Each knowledge rating generates a rerating of the source, which is calculated based on the results of (1), (2), (3) and (4), leading to source rerating, summarized in (5). To do this, we take into account the source's original confidence rating, the source's rating of his or her own knowledge, and the average knowledge rating by all users, without losing sight of the fact that the source is a qualified person and his or her opinion should be represented in the rerating. We generate a rule of three, Equation (6), where the unknown is a new source rating value (with respect to evaluations) based on the source rating, source's knowledge evaluation and the average knowledge evaluation.

$$\text{RevSources(y)} = (\text{Conf(y)} \times \text{C(xi->y))}/\text{C(y)} \tag{6}$$

### 3.9. Comparative of Existing Knowledge Management Models

Table 1 classifies the different knowledge management models according to the taxonomy defined by Kakabadse, Kakabadse and Kouzmin [25] as theoretical (primarily philosophical, conceptual and theoretical approaches), cognitive (examining cause-effect relations involving an optimization of the use of knowledge generally applied to industry), social network models (aiming to explain the

generation and transfer of knowledge in business environments by means of social networks and business practices), scientific and technological knowledge management models and hybrid models (that do not have the features of the models specified above or contain more than one characteristic of the above models). This table indicates the differences between each of the models and the GESTAC model.

GESTAC is a hybrid model that includes the characteristics of several models: it is theoretical because it adopts the philosophy of Nonaka's model with respect to knowledge conversion processes and applies the knowledge management life cycle. It is also a cognitive model, because it considers the knowledge rating processes taking into account several items. It has the features of a scientific/technological model because it analyses the tacit knowledge conversion processes by means of a prototype designed and implemented by a multiagent system.

**Table 1.** Comparison of GESTAC and other knowledge management (KM) models.

| Model Author | Main Focus and Characteristics | Difference to GESTAC |
|---|---|---|
| Nonaka | Theoretical, philosophical knowledge management | GESTAC includes a prototype |
| Wiig | Theoretical, philosophical knowledge management | GESTAC includes a prototype |
| Al-Mutawah | Cognitive (use: manufacturing) | • Covers multiple domains<br>• More general but less specific and powerful for knowledge management |
| Okike, Fernandes and Xiong | Cognitive (use: experience rating) | • Covers multiple domains<br>• More general but less specific and powerful for knowledge management |
| Gal | Theoretical and cognitive | Does not account for interpersonal relationships |
| Seidler, de Alwis and Gemunden | Technological and scientific | Does not deal with outsourcing |
| Brigui-Chtioui and Saad | Technological and scientific | Focused on explicit knowledge, not applicable to tacit knowledge |
| Karunananda, Ratnayke and Mendis | Technological and scientific | Implemented in Visual Basic and with input limitations |

## 4. Gestac Design

The incident management methodology is implemented according to ITIL as shows in Figure 4, (Source: ITIL and GESTAC): incident input (GESTAC users enter the incident in the computer system), incident transfer to the input desk (the computer automatically sends the incident to the incident area in GESTAC), incident logging, incident investigation and incident assignment in the KB, incident resolution and incident closure, although the incident could be reprocessed if better sources or solutions are discovered. If the incident is not resolved, the system awaits the resolution, and the incident is logged as open with no associated knowledge. Figure 4 shows the treatment of the incident component, following the ITIL logic. Figure 4 illustrates the processes related to incidents for application users to follow.

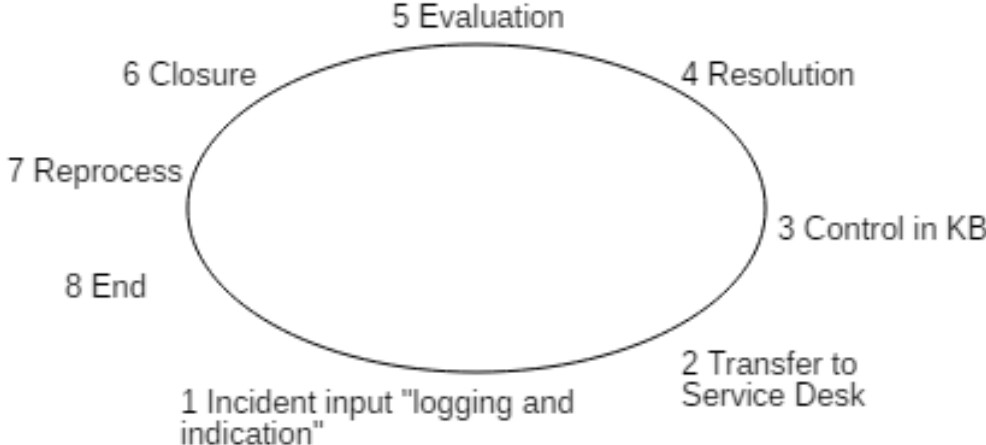

**Figure 4.** Incident management according to ITIL.

All this information is accessible for users. Figure 5 shows the key activities developed to rate the sources and knowledge. The participation of autonomous intelligent agents renders GESTAC highly independent of human intervention, making the model easily adaptable to changing environments.

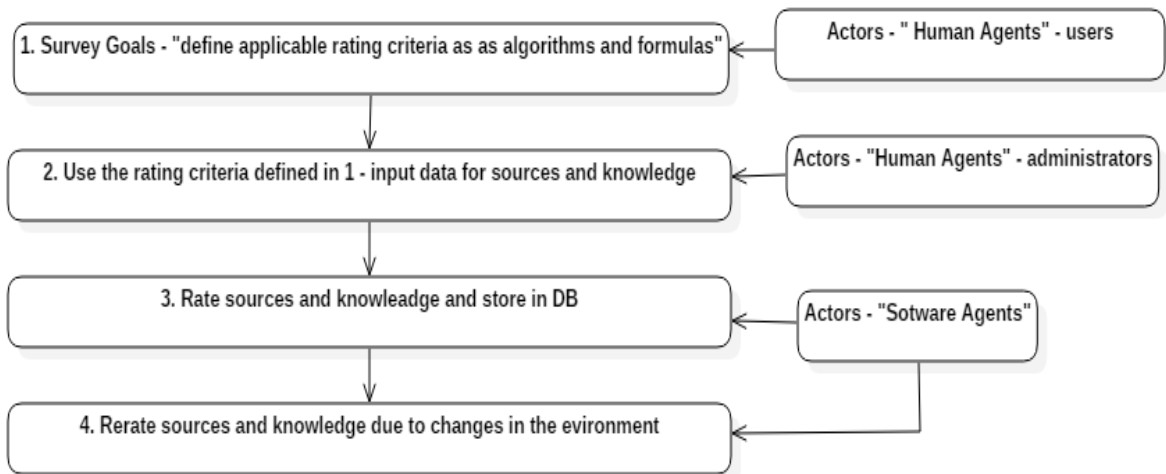

**Figure 5.** Key activities developed to rate sources and knowledge.

### 4.1. Relationship between the GESTAC Model and Knowledge Management (KM)

The GESTAC functional design is divided into three interactive layers (Figure 6): operational layer (input: incidents, areas, potential sources-> output: incident logging, areas, potential sources and specified knowledge), deliberative layer (transformation of potential into real sources, rating of captured knowledge, rerating of sources and knowledge) and informative layer (on-demand or other interfaces between agents). Figure 6 shows that the GESTAC model complies with all tacit knowledge management phases. The tasks of tacit knowledge identification, capture and storage are carried out in the deliberative layer. The distribution of captured and registered knowledge is carried out in the informative layer. The operative layer is designed for data input used by both the deliberative and the informative layers (i.e., to enter company areas, sources, incidents). This layer is associated with potential problem-solving knowledge, which is one of the interesting innovations of the GESTAC model.

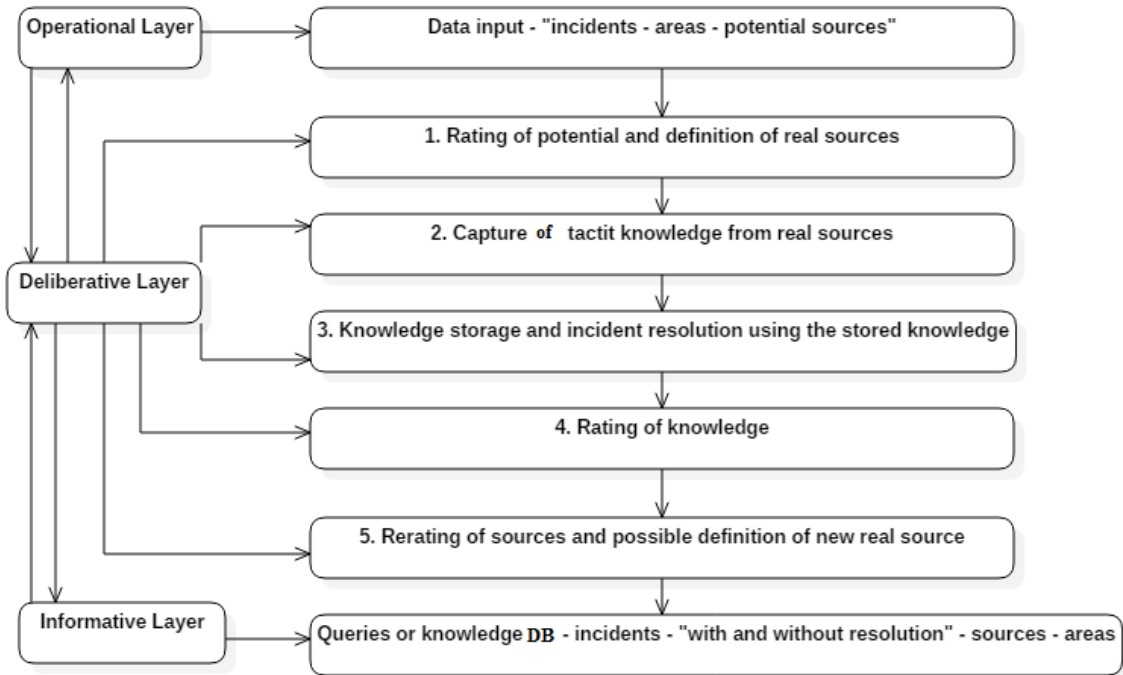

**Figure 6.** GESTAC layers.

*4.2. Design Decisions: Rating Item Definition*

Doucet and Sloep [50] formulated mathematical models for dynamic processes, mainly in the biology domain, and developed, on their basis, concepts applicable in any area. One such concept is that a model is verifiable if it is applicable, for which purpose it must demonstrate its connection with reality. According to Law and Kelton [51], a model can be validated qualitatively, gathering the opinion of human experts with respect to whether the model is adequate for the system it represents. Dee [52] describes different ways of validating models depending on their objectives. Validation of the conceptual model qualitatively evaluates whether the model represents reality. Validation of the algorithmic implementation analyses whether the mathematical model, expressed mainly by means of formulas, contributes to the fulfillment of objectives. Finally, validation of the software implementation evaluates whether the mathematical model is implemented according to its goals depending on the situations, processes and events to which the model refers. Based on this, we gathered expert opinions at design time in order to define the source and knowledge weighting items based on business realities.

We designed two requirements to be met by respondents: (1) hold or have held knowledge management responsibilities: (a) by personally pushing forward the issue and, (b) by answering questions of an email survey regarding rating sources of knowledge; and (2) how to specify tacit knowledge. The respondents were allowed to choose more than one item per question, some of which were proposed by the researchers and others by respondents (item specified as others, such as common sense, method, definite interest in the area, hardworking, tidy, etc.). The results of the surveys led to the model formulas.

Briefly, Figures 2–6 illustrate the aspects of the GESTAC model that were analyzed in the previous sections. First, the model as a whole is composed of interactive components. Second, evaluations of sources and knowledge are dynamic. Third, the incident resolution processes are taken from ITIL. Fourth, the activities of the model necessary to meet the specific objectives are related to their respective actors. Finally, the GESTAC model is structured by layers according to the tact knowledge management phase objectives.

## 5. GESTAC_APP-GESTAC Model Prototype

This section presents a specific prototype of the GESTAC model. It was originally intended to be used in a specific situation and domain, with the possibility of extension to other business domains, subject to changes to the initial data.

The main aim of this prototype is to validate the GESTAC model through experimentation. Its purpose is to represent the processes, omitting other attributes like security, usability, visual design, scalability and performance, which are left for future extensions. The idea was to separate the business logic from the design logic in an attempt to achieve easily understandable code to improve maintainability, increase coupling to facilitate error correction and include improvements by experts in distributed environments by working with multiple real-world agents. To do this, we followed recommendations by Hutchin [53], who offered a new perspective on cognitive science by adopting the proposed methodologies.

One innovation of GESTAC_APP is the use of software agents integrated into a multiagent system for managing tacit knowledge in a specific domain.

### 5.1. Architecture

The GESTAC prototype was implemented as a web application in order to achieve the model's goal of sharing knowledge across different users from different places within a business domain. Figure 7 shows a diagram of the GESTAC prototype architecture, and Figure 8 the back-end design. The front-end (Figure 7) should act as the interface with the previously logged on user, where the browser is located on the client device. The back-end contains the services that are used by the front-end to perform functions, including data processing, data storage and system logic. The front-end and back-end are connected by means of a network that may be located on the same device (when the front-end and back-end are together) or a local area network (via intranet in connections at the same company) or via the Internet (in wide area connections). As shown in Figure 8, we used the criterion of separation into three layers for the back-end design: web layer, responsible for all the services, domain layer, responsible for the operational logic, and the data storage access layer. The WEB module contains the presentation and business layers. The presentation layer has all the application's front-end web behavior. It uses web pages that were implemented in JSP (JavaServer Pages), HTML (Hypertext Markup Language), CSS (Cascading Style Sheets), JavaScript, AJAX (a set of web development techniques using many web technologies on the client side to create asynchronous web applications). The web module business layer was implemented using servlets to respond to the requests launched from the presentation layer using web pages. The business logic is executed, and the data access objects (DAO) in the persistence layer are used by the persistent entities to communicate with the database. Filters were used to filter the authorization of the requests from the front-end and web resources when the user is not logged on, controlling that the only access point is the login page. Each request is mapped to the servlet that processes and executes the logic specific to the action requested for the web page. The interface, where Hybernate will be used for data persistence, defines the basic administration methods and a class implementing the methods.

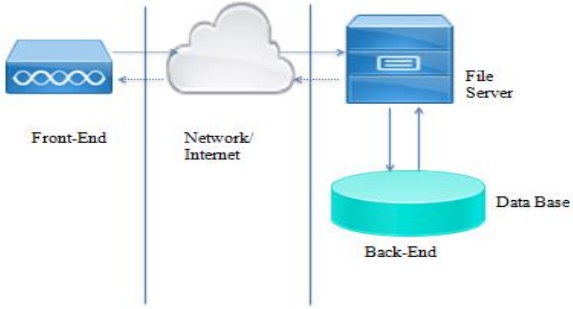

**Figure 7.** GESTAC prototype architecture.

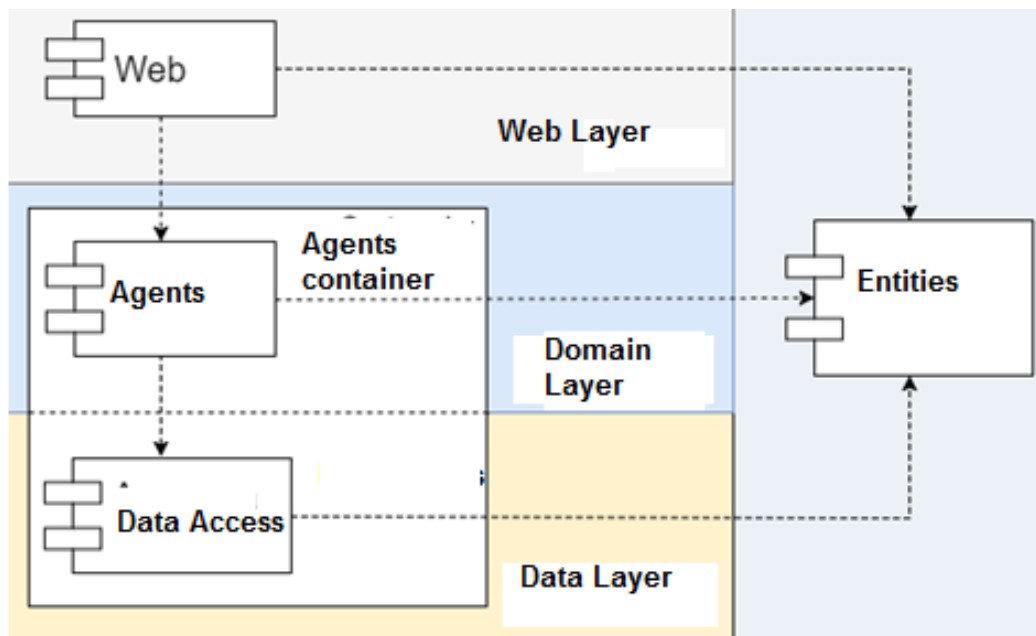

**Figure 8.** Back-end architecture.

### 5.2. Rationale for the Use of the Multiagent Systems Paradigm in the GESTAC Prototype

The prototype was implemented using the multiagent systems paradigm, essentially designed for distributed environments to emulate the behavior of knowledge in GESTAC. Multiagent systems reduce cognitive overload by sharing contexts and grouping actors that have interests, roles and tasks. GESTAC groups incident management by business areas. Agent proactivity facilitates adaptability to the changes in GESTAC environments (for example, new sources and knowledge), which are processed automatically by software agents without human agent intervention. As a result, the environmental changes are automatically reflected in the data. As the features are independent of the logic, the responses to changes and data updates are separate from the underlying logic. For example, the knowledge rating functionality is not acquainted with the output rating or update; this is the domain of a specialized agent created for this purpose. As a result, the logical changes do not affect the updating process, thereby minimizing the likelihood of bugs and assuring that new functionalities do not have to be concerned with knowledge rating. This separates the data processing functionality from the data generated using the data. Another advantage of this feature is the possibility of future extension. For instance, if another future system were to have an impact on the GESTAC database and led to new ratings, they would be taken into account without having to make functional changes to the model. The agent rationality feature means that each agent specializes in a field and is capable of getting the best solution according to its own rules, leading to a modular design of GESTAC. Agent learning means that each agent always has the up-to-date knowledge that it needs, for example, they know which are the best sources and knowledge at any time. In general, other forms of modular organization of the system could have been used instead of using agents, but in the early stages of the project it could be verified that the prototypes developed with agents were ahead of the use of other technologies: they offered better results in terms of maintainability, computational efficiency and adaptability in contexts of changing information like this one. Thus, the intervention of expert domain users was minimized, thanks to the flexible, reactive, proactive and social behavior of the agents.

### 5.3. GESTAC Prototype Agents: Some Examples of Algorithms and Their Encodings

Software agents were used in the domain and data layers, on which ground it was decided that they should all be part of the same container in order to simplify the communication overload. Agents were used to represent the KB and promote model independence, extensibility, maintainability

and performance. Accordingly, more than one agent can access the KB concurrently, as there is more than one instance for each agent. To do this, we used a generic agent for the least common operations and a specific agent for each of the key entities, for example, the SourceDBAgent for sources and the KnowledgeDBAgent for knowledge.

Figure 9 illustrates an example of the coding to achieve the communication among software agents by means of ACL messages. The SourceDBAgent receives the login and password from whoever wants to log in as source, sends a generic ACL message containing the received parameters, the message-emitting agent identifier and a function in order to instantiate the source object. The SourceDBAgent waits for the response to the sent message, using a blocking method according to which it ignores any other behavior until it receives new messages. When it receives the response, the behavior that it was blocking is resumed. The blocking logic would be as shown in Figure 10.

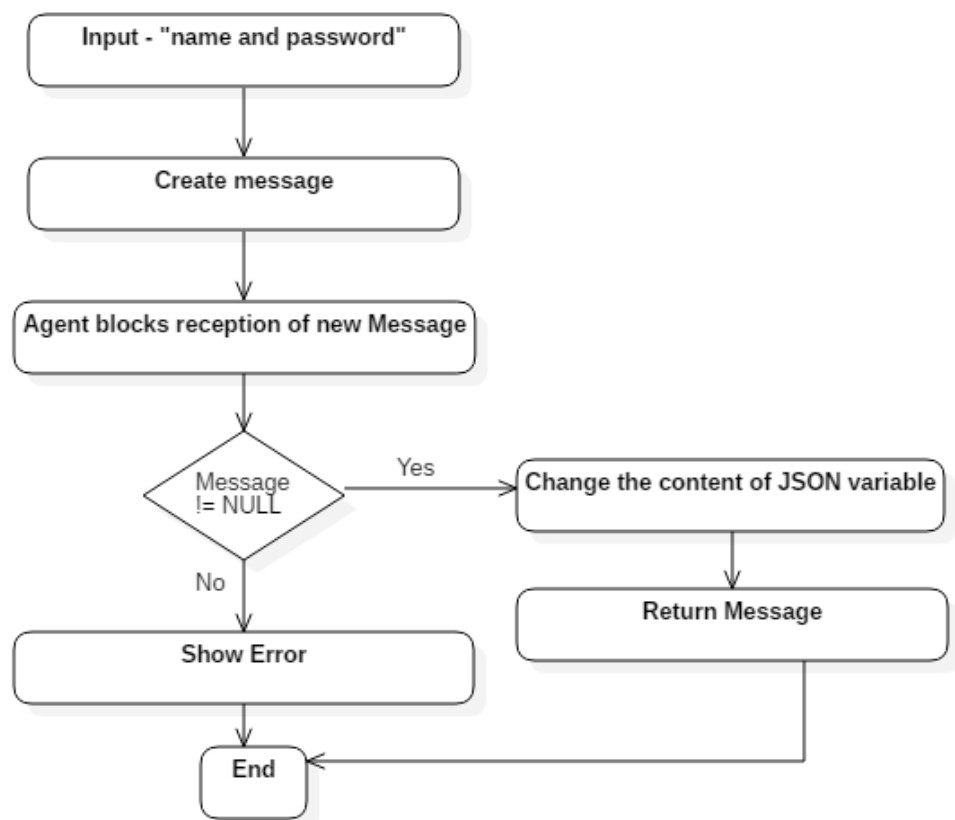

**Figure 9.** The logic of algorithm of sources input.

It was encoded like this:
```
    public void action(){
    Message message = receive()
if (message!=null) {   //Message received, is processed.
        } else {
        block();
    }
    }
```
The source input operations follow the logic shown in Figure 9, where checks are run, as shown in Algorithm 1, to verify that the source does not exist as follows:

"requestedAction&parameter1&parameter2"
"requestedAction&{field1:parameter1,field2:parameter2}"

The procedure for source input follows the logic shown in Algorithm 2. If the source input operation is successful, the data gathered are mapped to and read as a JSON string, as shown in Algorithm 3.

---

**Algorithm 1.** Check for existing sources

---

Function NonExistentSource
Input: Source Name
Output: True (if source exists) or False (if source does not exist)
1. CreateACLMessage (name of agent that processed the operation, agent_operation (), name of source)
2. Send Message ()
3. Agent blocks the reception of new messages
If message answer! NULL
Return True
Else
End

---

---

**Algorithm 2.** Log sources

---

Function NonExistentSource
Input: Source Name and password
Start
Output: True(if source exists) or False
1. Creates message with parameter name and password
2. Agent blocks reception of new messages
If message! NULL
Return message
Else
Error
End

---

---

**Algorithm 3.** Save source input

---

1. Messages are created containing the agent that will process the operation with the source and operating mode parameter data
2. A JSON message maps the specific source and area data
3. The message is sent
4.The agent receiving the message blocks the reception of new messages.
If agent receives message! NULL
Save Source
Else
Return   False
End

---

We also used ACL (Agent Communication Layer) messages and JSON (JavaScript Object Notation) strings. ACL messages werComparison of GESTAC and other knowledgee used to check that a source exists and evaluate and save source data. The text between the start of the content and the first occurrence of "&" indicates the action that is requested of the agent. In practice, it acts like a method name. The following parts of the message are the data required to take that action. They can be regarded as parameters, entered as text if they are simple values or as a JSON string if they are objects. JSON was selected as a means of serialization in order to exchange and store data on several grounds: it is flexible and easily generalizable, it can generate an output of limited size, it is in widespread use and is compatible with any data structure, and it can be used with any technology. All of the above issues simplify data exchange with respect to XML (Extensible Markup Language).

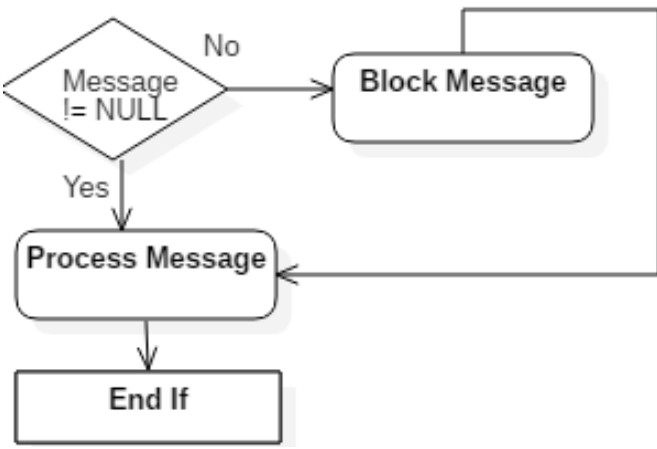

**Figure 10.** Blocking logic.

The sources described in Section 3.8, and calculated using Equation (6), are rated and rerated using a function that returns the final score of each source at any time. The parameters that are passed to the function are the source entity and the result of the knowledge assessments. The source rerating is given by the knowledge evaluation by the source multiplied by the average rating of the knowledge entered by the users, divided by the evaluation of the knowledge made initially by the source.

Figure 11 shows the Save sources logic, Algorithm 3.

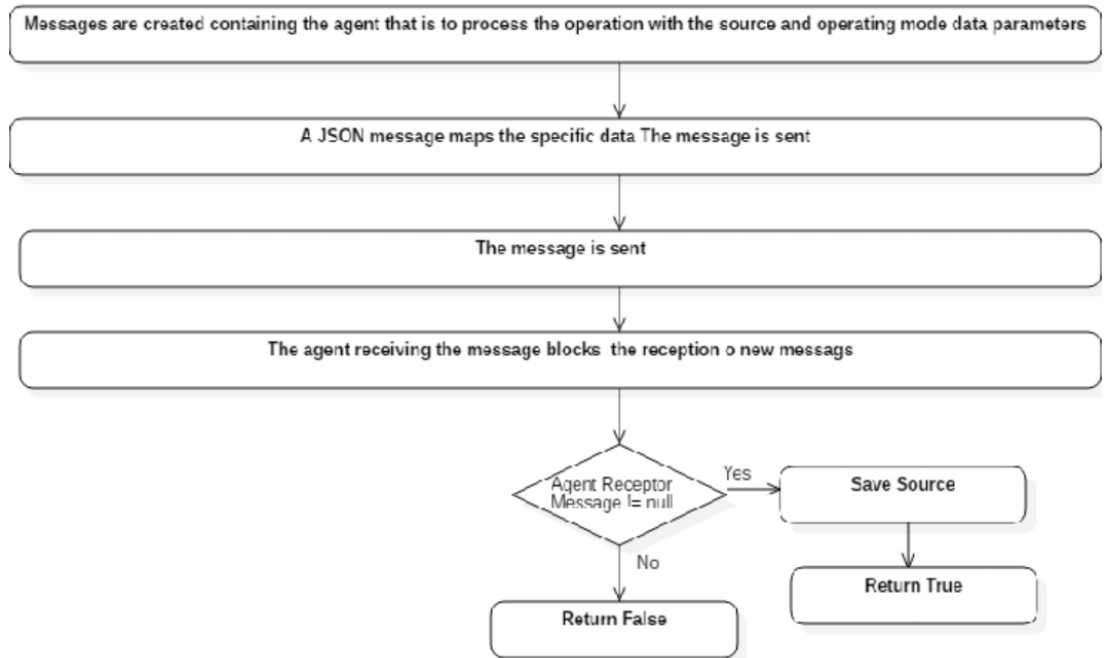

**Figure 11.** Save source logic.

Once these three algorithms have been presented, it is possible to obtain information on examples of behaviour, data, messages and real system executions, through the content available in [37].

## 5.4. The Main GESTAC Logic Implementation Agents

**InterfaceAgent**. This is the only agent known to the web drivers and complies with the *Interface* design pattern. The rest of the system uses this agent to access and receive responses from all agent services. It is responsible for carrying out maintenance tasks, like simple entity creation and deletion. Accordingly, the other agents do not have to be unnecessarily complex. It receives all the requests from

the top layer, which are then derived to the respective agent or the bottom layer. InterfaceAgent is an extension of the GuiAgent, which is a special agent type provided by Jade. In this case, it is used to indicate that this agent is the access point to all other agents. There may be more than one instance of some of these agents at any given time.

**IssueSearchAgent**. This agent is responsible for searching for user issues. To do this, it collaborates with the IssueAgent, requesting searches and unifying and processing the results.

**IssueAgent**. The role of this agent is to resolve some of the user issue searches, as requested by the IssueSearchAgent. It is an on-demand agent.

**IssueManagementAgent**. This agent is responsible for performing issue-related tasks other than user search.

**KnowledgeAgent**. This agent is responsible for enabling database knowledge aggregation and its respective ratings, but does not take charge of the evaluation processes. It searches for the best knowledge of a problem.

**KnowledgeScoreAgent**. This agent takes charge of knowledge reratings, detecting new situations and recalculating the score in the event of new instances, alerting the respective source in the event of changes to be entered. It is responsible for updating the best knowledge for each incident but not for updating its source.

**SourceAgent**. This agent is merely responsible for getting the area of the best source for each incident.

**SourceScoreAgent.** It plays a similar role to the KnowledgeScoreAgent with respect to sources. It takes charge of updating source rating, and saves the best rated source for each area.

Figure 12 shows agent interaction.

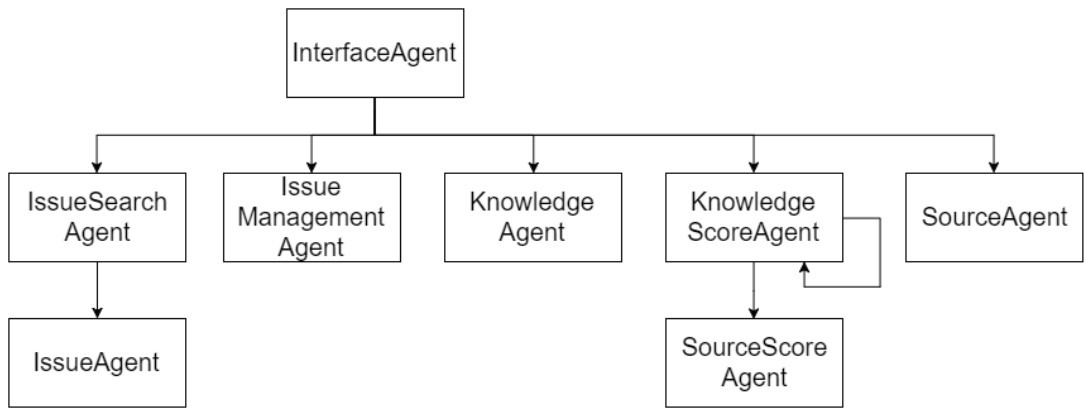

**Figure 12.** Agent Interaction.

### 5.5. WEB Layer

The web component is the only interface with the client process (browser) and is composed exclusively of web drivers that implement the above services. These services use HTTP methods with the following criteria:

GET: queries data.

POST: sends data to server, creates new agents.

PUT: updates existing entities, used exclusively for the source editing functionality.

DELETE: removes entities.

### 5.6. Domain and Data Access Layers

The domain and data access layers are where agent use is concentrated. The InterfaceAgent has normal methods without any additional behavior and is used from web layer classes. All agents, except for SourceScoreAgent and KnowledgeScoreAgent, have a single behavior. Each SourceScoreAgent

instance has a single behavior that is decided depending on an agent builder parameter depending on agent function. Any given KnowlegeScoreAgent instance can have one behavior, whereas another may have two behaviors simultaneously. Behaviors are used to meet the need to maintain the best knowledge for each area. They are initiated at application launch. Therefore, they do not require external intervention to perform their task. Figure 13 shows the model of the agents used in the domain layer. For KB access, a generic agent was used for less common operations, and specific agents were employed for each of the key entities (Figure 14): IssueDBAgent for problems, SourceDBAgent for sources. We decided on this separation because, being the key entities, they are the ones that are most used in the system and, therefore, call for more frequent operations. This provides for concurrent operations not available with a single agent. Figure 15 shows the GESTAC prototype class diagram. We decided to assign a field called ID to identify each entity. This field is used as the database primary key in order to rule out reference handling problems. Although the Issue class includes a Subject and Incident list, for simplicity's sake, only one item can be added to these lists in this implementation. The idea of using lists is to lay the groundwork for future work on this implementation, as a possible first step would be for a problem to have more than one subject and incident. The Issue-BestKnowledge and the AreaBestSource classes are used to represent the best knowledge for an incident and the best source for an area, respectively. They are necessary for the implementation, but are not, strictly speaking, model entities.

It is possible to consult concrete examples of instances of the class diagram present in Figure 15 in real scenarios such as that described in Section 6 through the on-line repository [37].

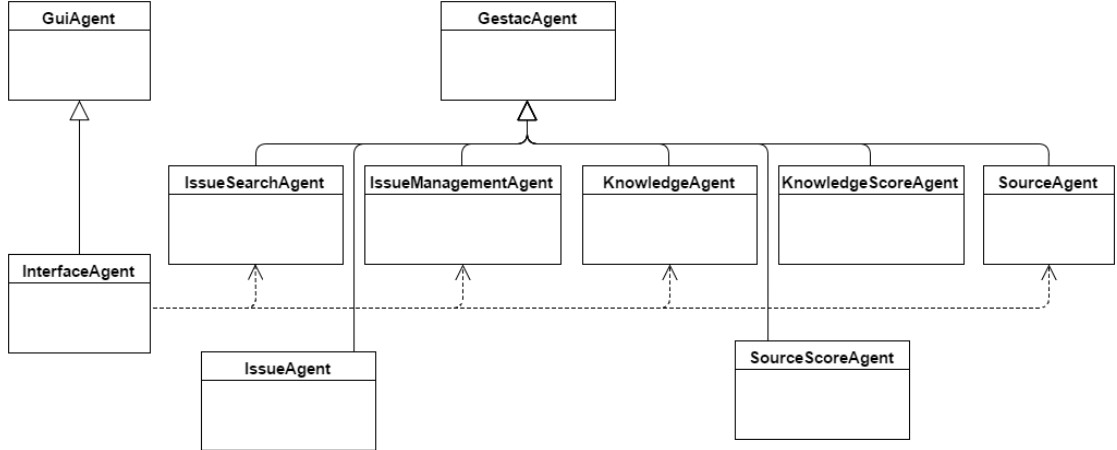

**Figure 13.** Domain layer agent model.

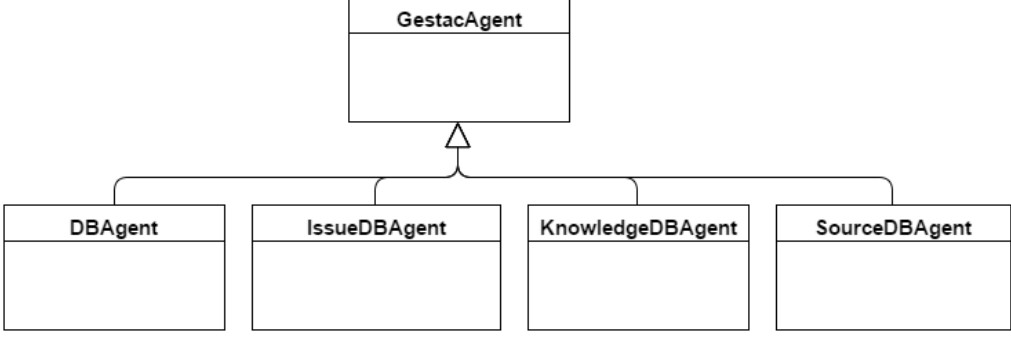

**Figure 14.** Data access layer agent model.

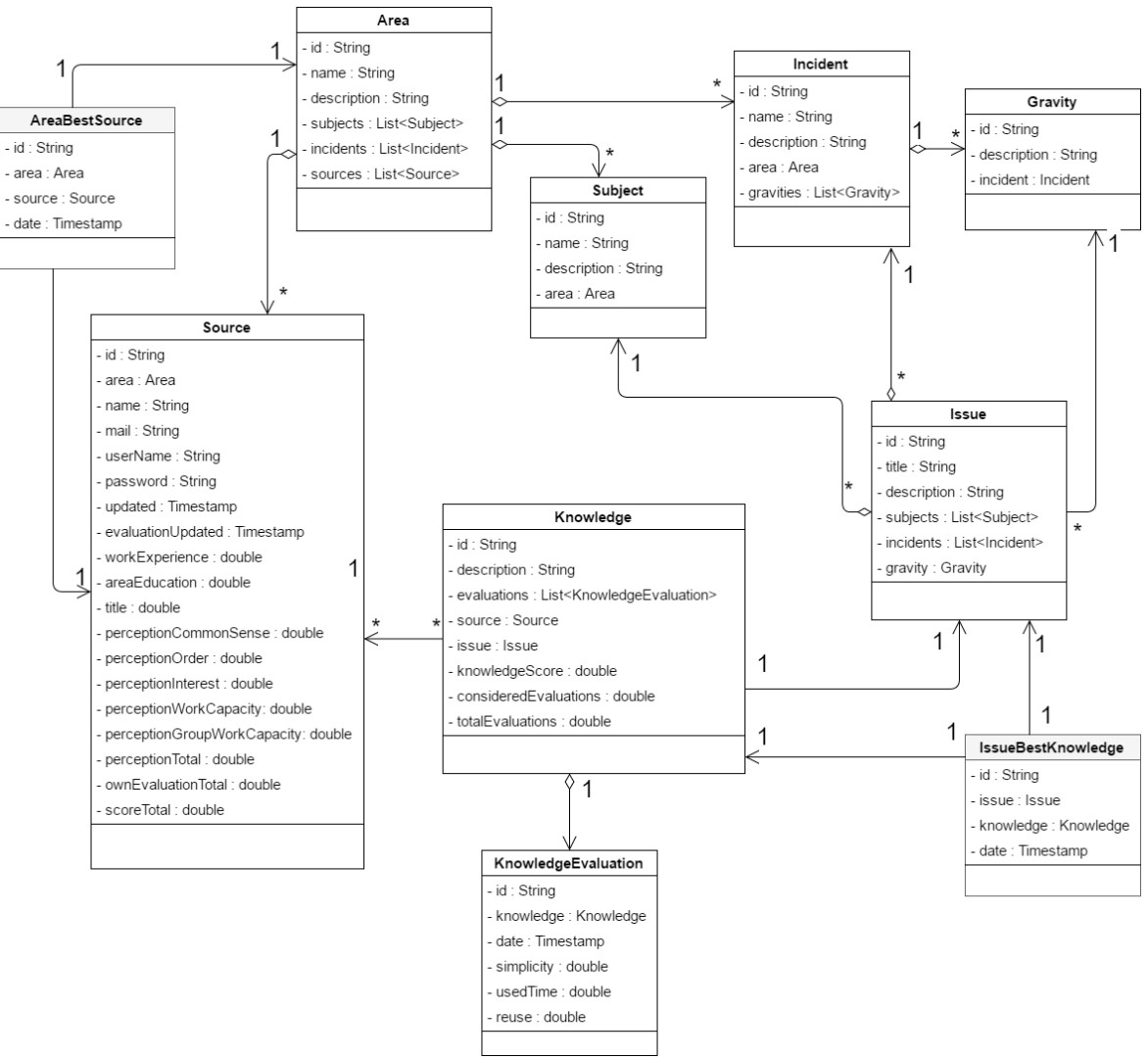

**Figure 15.** GESTAC prototype class diagram (where * is a cardinality from 1 to N).

## 6. Experimentation

The aim of this experiment is to empirically test the originally stated hypothesis that: by applying tacit knowledge management, GESTAC model could have a positive impact on the resolution of business incidents logged by users in a specific domain by speeding up incident resolution times compared with the same domain where the model was not used. The experiment consisted of comparing incident resolution times across two groups of engineers that differed with respect to whether or not they used GESTAC.

### 6.1. Theoretical Groundwork of GESTAC Experiment

According to Kitchenham [54], the phases of the experimentation cycle are: design, data collection, analysis, observation and results reporting. According to Juristo and Moreno [55], experimental design can be used to validate the beliefs and ideas of the experimenter by matching their suppositions, assumptions, speculations and beliefs with a specified reality. To do this, it is necessary to describe the variables by means of their relations, some of which (especially causal relations) can be explained by functions. As far as Salkind and Escalona [56] are concerned, any scientific experimentation implies processing one or more independent variables, whose value range generates the treatment level. Participant subjects must be assigned at random and have the same experience and conditions. The case is internally valid if the independent variable has effects on the dependent variable or variables

and externally valid if the results can be extended to other domains. The design proposed by Sato [57] considers that any experiment must have treatment groups and control groups.

According to Kerlinger [58], experimental design has two major goals: it should try to both answer the major research questions and measure variance.

### 6.2. GESTAC Experimental Design

We applied what Anscombe [59] termed sequential experimentation, leading to changing results, as the aim is to analyze the relations between the environment and the values to be researched. The factorial GESTAC experiment analyzed the interaction between two factors in real business scenarios. The experiment took place in the real world of a company.

The experimental design addressed the influence of the tacit knowledge management (independent variable) applying GESTAC on the incident resolution time (dependent variable), trying to quantify the standard deviation by means of variance. We tried to carry out what Pazos [60] refers to as a crucial, fine or critical experiment, where just a few experiments are sufficient to test or reject the hypothesis. The experiment was conducted at the IT department of the Educational Institute of the Child, which regularly receives and resolves incidents logged by the organization's users nationwide in the fields of support, networks and computer security, software, project management and hardware.

### 6.3. Experimentation Methodology

The experimental design was developed as follows:

(a) We took into account 10 incidents to be resolved from different IT areas.
(b) Two technicians for each area were asked to resolve the same incident: Technician A would have to resolve the incident without applying GESTAC and Technician B would do so using the system.
(c) Technicians A and B were distributed at random. If there was more than one problem per area, another lottery was held, and one and the same technician could be A for one issue and B for another. All the people were significant since they regularly resolve the incidents logged by users for each area.
(d) An attempt was made to assure that new incidents were logged in each area, and they were pretested by benchmark users from different business areas.
(e) For each response, we quantified the values of the incident resolution time dependent variable (considering the real lapse of time between when the technician starts to resolve the incident to when it is finally resolved, minus any breaks). The Group B technicians were given a user manual explaining how the model works.

### 6.4. Incidents Logged and Recorded in the System

Incidents: (1) Computer Networks and Security Area—the Internet connection is often unreliable. (2) Software Area—the electronic record cannot always be signed using a digital certificate. (3) Project Management Area—the tenders do not always contain the precise terms required by the applicable rules. (4) Hardware Area—the user often cannot print a document at the printer. (5) Support Area—user does not know how to get a digital certificate. (6) Networks Area—the network is often very slow from 13 to 15 h. (7) Computer Networks and Security Area—the user needs to block unnecessary applications. (8) Hardware Area—personal computer (PC) xx sometimes loses connectivity. (9) Computer Networks and Security Area–the user is connected to the Internet, but the connection to the application managed by personnel often drops out. (10) Hardware Area—the data center is not properly air conditioned.

### 6.5. Results and Statistical Tests

The results were gathered from a questionnaire and were analyzed as follows: the null hypothesis (H0) held if the resolution times (RT) used by the technicians in Group A were statistically similar to

the times used by technicians in Group B. If RT were less than for Group B, the alternative hypothesis (H1) could hold, on which grounds only two options would be considered: the resolution times were better for Group B than for Group A. We asked the technicians to record the resolution time for each incident on the time sheet (Table 2), which we used to calculate the mean and variance (Table 3). As a binomial distribution with only two possible values was used, the possibility of an improvement in the resolution times would be $p = 0.5$ for both options. The test statistic is the number of positive results for the comparison between the times of both groups, and neither the negative nor the equal results are considered. This is a one-tailed test, as H1 is only analyzed in one direction. We decided that the acceptable risk of test result error, that is, the significance value, would be 5%, meaning that the confidence level of the results is 95 %. Therefore, according to Student's *t* distribution table for the above values (Table 4), the critical value for 10 tests would be 1.812. This means that the region of rejection for H0 could be verified if the resulting statistic is less than 1.812.

**Table 2.** Resolution time by incident and technician.

| Incident | Resolution Time (RT), Minutes | Technician (A/B) |
|:---:|:---:|:---:|
| 1 | 120 | A |
|   | 30 | B |
| 2 | 105 | A |
|   | 45 | B |
| 3 | 180 | A |
|   | 105 | B |
| 4 | 45 | A |
|   | 30 | B |
| 5 | 80 | A |
|   | 50 | B |
| 6 | 50 | A |
|   | 45 | B |
| 7 | 60 | A |
|   | 60 | B |
| 8 | 55 | A |
|   | 20 | B |
| 9 | 20 | A |
|   | 15 | B |
| 10 | 55 | A |
|   | 35 | B |

**Table 3.** Resolution time mean and variance by technician groups.

| Technicians | Resolution Time | Mean | Variance |
|:---:|:---:|:---:|:---:|
| A | 770 | 77 | 1849 |
| B | 465 | 46.5 | 530.22 |

**Table 4.** Student's table entry for 10 tests with a significance level of 0.05 and 0.1 (df 10).

| 0.10 | 0.05 | 0.025 | 0.01 | 0.05 | 0.0005 |
|:---:|:---:|:---:|:---:|:---:|:---:|
| 1.372 | 1.812 | 2.228 | 2.764 | 3.169 | 4.587 |

According to Mason [61], it would be recommendable in small tests like this to use two values. On this ground, we also considered an error margin of 0.10 for the analysis. In this case, according to Student's *t* distribution table (Table 4), the critical number would be 1.372.

For the calculation of resolution time, mean and variance, the basic statistics have simply been obtained from the data collected in Table 2 for both groups (A and B). From the data shown in Table 3, we can conclude that the standard deviation for Group A (43) is greater than for Group B (23.026). Thanks to the use of Student's *t*, it is intended to check whether the average difference in the resolution times obtained in Table 3 is statistically significant or not. For this purpose, a study was carried out with a significance level of 0.05 and 0.1, shown in Table 4. We used the data from Table 4 to calculate the value of the statistic (t) for comparison with the critical value, depending on the significance level (1.812 and 1.372 with a significance level of 0.05 and 0.10, respectively). As recommended by Mason et al. [61], we analyzed the results taking into account two significance values. We have two independent non-parametric samples with different variances (where x1 is the mean for Group A and x2 is the mean for Group B, and s1 is the standard deviation for Group A and s2 is the standard deviation for Group B). Equation (7) is used to calculate the value of the significance statistic, because we have two independent non-parametric samples with different variances. Applying this formula, we found that $p = 1.52$. On this ground, H1 would hold considering a significance level of 0.05, as the *t*-value, 1.812 (Table 5), is greater than the result for the *p*-value. Considering the 0.10 significance level, however, H1 would not hold because 1.52 is greater than 1.372. Figure 16 shows the areas for which there is and there is not any evidence to reject H0, according to *p*-value.

$$p = (x1 - x2)/(s1 - s2) \tag{7}$$

**Table 5.** Comparison of significance level, *p*-value with unchanging probability of rejection.

| Level of Significance | *t*-Values | Probability of H0 Rejection |
|:---:|:---:|:---:|
| 0.1 | 1.372 | 1.52 |
| **0.05** | **1.812** | **1.52** |
| 0.025 | 2.228 | 1.5 |

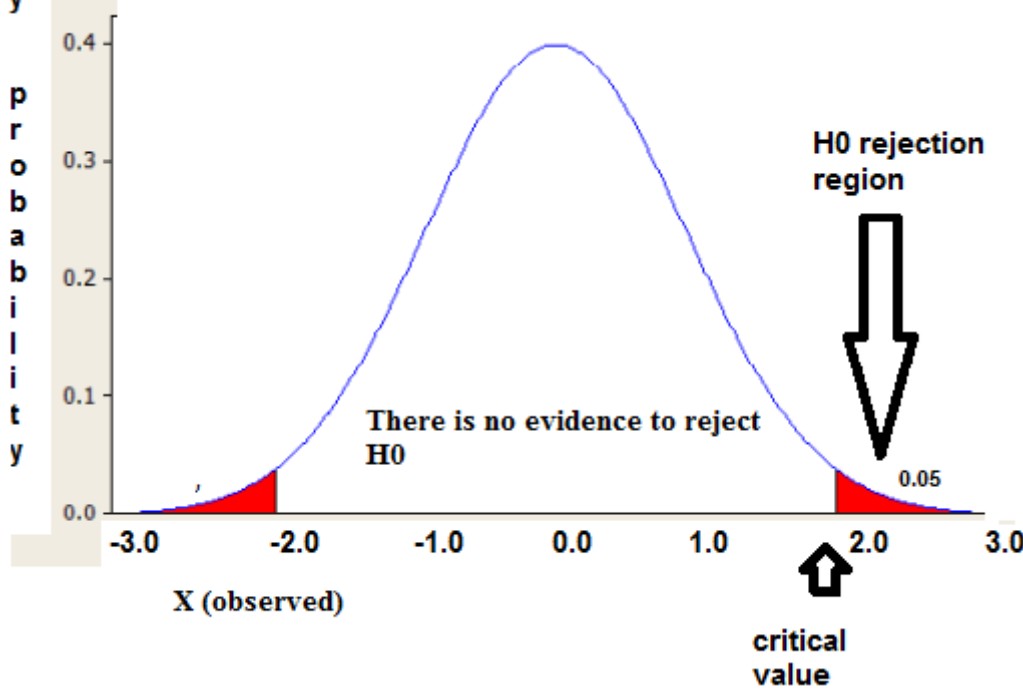

**Figure 16.** H0 rejection area according to *p*-value.

Table 5 illustrates the decision values for the study carried out, in such a way that in the light of the data obtained the initial hypothesis must be rejected (groups A and B had equivalent results) and the

hypothesis that the users who used GESTAC spent significantly less time must be accepted. Figure 17 compares the statistical values resulting from applying Equation (7) with the *p*-value and level of significance according to the Student's *t*-test illustrated in Table 5, if the value of t was unchanged for 10 tests (source: Minitab [62] with GESTAC data).

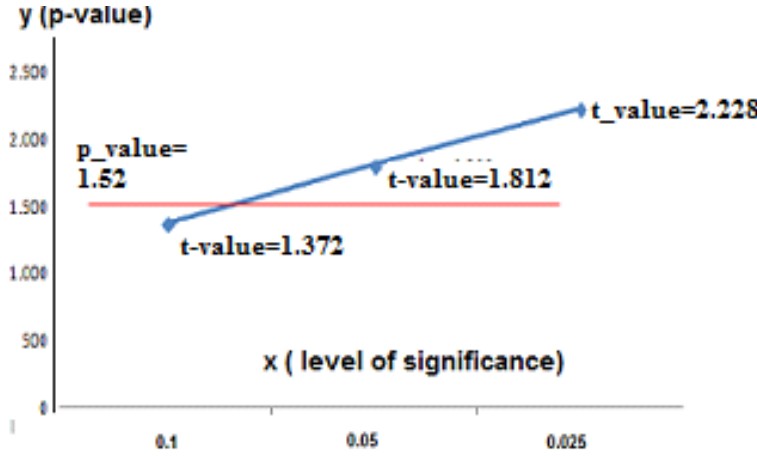

**Figure 17.** Graph of statistical values plotted from Table 5.

We counted the positive differences between Groups A ($\mu$t1) and B ($\mu$t2) resolution times (Table 6), taking the differences between resolution times specified in Table 1.

**Table 6.** Data according to difference sign.

| Incidents | Sign |
| --- | --- |
| 1 | + |
| 2 | + |
| 3 | + |
| 4 | + |
| 5 | + |
| 6 | + |
| 7 | |
| 8 | + |
| 9 | + |
| 10 | + |

We did not take into account 0 results. If $\mu$tA $\neq$ $\mu$tB, the null hypothesis (H0) was rejected. The statistical summary and its interpretation are shown in Table 7.

**Table 7.** Summary statistics.

| Variable | Observations | Minimum | Maximum | Mean | Standard Deviation |
| --- | --- | --- | --- | --- | --- |
| 120 | 9 | 20.000 | 180.00 | 72.222 | 46.711 |
| 30 | 9 | 15.000 | 105.00 | 45.000 | 26.693 |

Sign test/Two-tailed test:

| | |
| --- | --- |
| N+ | 8 |
| Expected value | 4.000 |
| Variance | 2.000 |
| *p*-value | 0.008 |
| alpha | 0.05 |

The *p*-value is computed using an exact method.

Test interpretation:

H0: the two samples follow the same distribution

H1: the distribution of the two samples is different.

As the computed *p*-value is lower than the *t*-value at the significance level alpha = 0.10, the null hypothesis H0 should be rejected, and the alternative hypothesis H1, accepted. The risk of rejecting the null hypothesis H0 when it is true is less than 0.78%.

On this ground, we can infer that H1 held and H0 was rejected. To check this, we can apply Equation (8) to the data to gather the value of the test statistic (Z), where x is the maximum number of positive data items and n is the total number of data. Equation (8) is used to calculate the critical value for the sign test. As Z is greater than the critical value shown in Table 5 for 0.05, we can infer that the data are statistically significant.

$$Z = (2x - n)/(\sqrt{n}) \tag{8}$$

The decision rule taken was that if the number of positive signs exceeds the critical number, H0 could be rejected. Looking at only 4 out of the first 6 data shown in Table 6, the relationship between the critical values calculated by applying sign tests and the number of positive data is shown in Table 8. Table 8 includes a statistical study for the sign test. This study is carried out to validate the experiment, and to verify that the incidents managed by both groups were equivalent in typology and type of resolution. Figure 18 shows the statistical value for the sign test depending on the number of positive data.

**Table 8.** Statistical value for sign test depending on number of positive data with significance level 0.05.

| Statistical Value for Sign Test | Positive Data | Total Data |
|:---:|:---:|:---:|
| 2.449 | 6 | 6 |
| 1.63 | 5 | 6 |
| 0.81 | 4 | 6 |
| 0 | 3 | 6 |

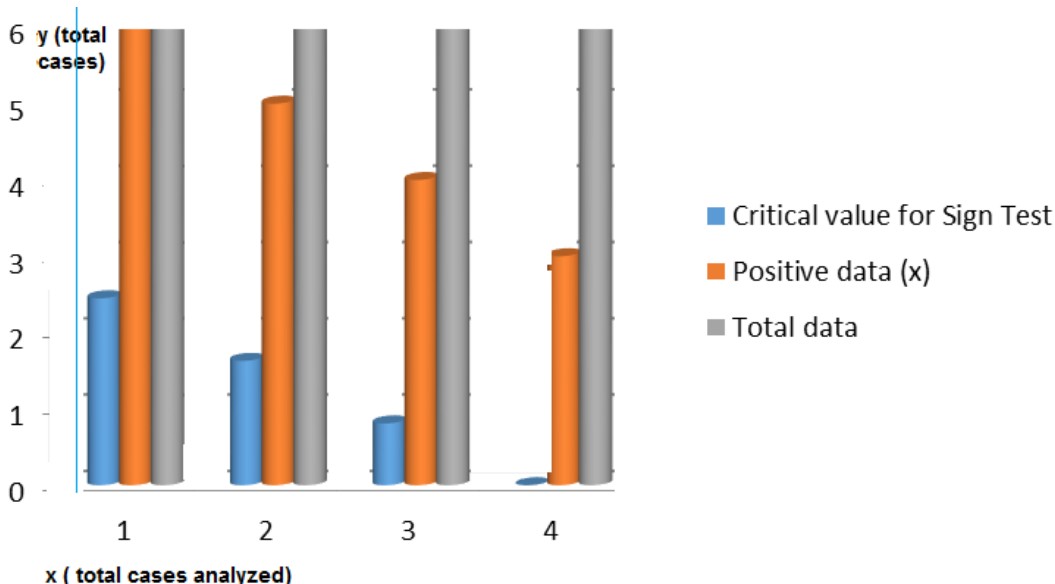

**Figure 18.** Statistical value for sign test.

The results shown in the Table 8 can be interpreted as follows: if the total number of sample data is unchanged and the number of positive differences between Group B and Group A resolution times decreases, the critical number would decrease to 0.

### 6.6. Mann–Whitney U Test or Wilcoxon Rank Sum Test

This test evaluates frequency distributions (the U-statistic) by comparing the medians of Group A with Group B. If the U-statistic for both groups is identical or the *p*-value is greater than the critical number used (0.05), the H0 cannot be rejected. The Mann–Whitney test uses Equation (9) to calculate the $U(x)$ statistic defined as the minimum U for A and B and calculates the value range including the differences of medians for both groups, where n is the size of each group and R(x) is the sum of the position of the ranges of each group (A and B, with x = 1 for Group A and x = 2 for Group B). We ran the rgui (R – graphical user interface) program to conduct the Mann–Whitney U tests [52] using the data from Table 1 for Groups A and B.

$$U(x) = n1n2 + ((nx(nx+1))/2 - \sum Rx \qquad (9)$$

Applying Equation (9), the lower value is used to search the Figure 19. Since U1 is the minst value (U1 = 120 – 60 = 65), we look for U1 in the Figure 19 and we find that, with a critical value of 0.5, the *p*-value is 0.02974. This is less than the *p*-value (0.05), thus indicating the probability of rejection of H0.

―――――――――――――――――――

➢   GROUP_A=c(120,105,180,45,80,50,55,20,55)

➢   GROUP_B=c(30,45,105,30,50,45,20,15,35)

➢   Wilcox.test(GROUP_A,GROUP_B)

Wilcoxon rank sum test with continuity correction

Data:Group_A and Group_B

W = 65.5,p-value = 0.02974

Alternative hypothesis:true location shift is not equal to 0

―――――――――――――――――――

**Figure 19.** Mann–Whitney test by rgui.

If the critical value used was 0.10, a value of up to 16 would be acceptable for the H0 not be rejected. The Mann–Whitney test was run using two applications—rgui and XLSTAT—because both returned an approximate *p*-value value.

The results of the one-tailed Mann–Whitney test (Figure 19) can be interpreted as follows: the H0 can be rejected as the *p*-value (number denoting the acceptable error possibilities), 0.02974, is less than the critical value (0.05). The resulting data confirm that the population distribution in both groups is different. On this ground, we can infer, according to the applied statistical research criteria, that the model used variables yielding statistically significant results, and H1 holds. This means that the incident resolution times for Group B technicians who used GESTAC were lower than for Group A technicians who did not use the model.

The results yielded by 10 Mann–Whitney tests are as follows:

Wilcoxon signed-rank/test (two-tailed test):

| | |
|---|---|
| V (standard variance) | 2.524 |
| Expected value | 18.000 |
| Variance(V) | 50.875 |
| *p*-value | 0.012 |
| alpha | 0.05 |

Test interpretation:

H0 = The two samples follow the same distribution

H1 = The distributions of the two samples are different

As the computed *p*-value is lower than the significance level alpha = 0.05, the null hypothesis H0 should be rejected and the alternative hypothesis H1, accepted.

The risk of the null hypothesis H0 being rejected when it is true is less than 1.16 %

If 5 instead of 10 tests are considered, the results are different, as follows:

| | |
|---|---|
| V. | 10 |
| V(standard variance) | 2.524 |
| Expected value | 5.000 |
| Variance(V) | 7.500 |
| *p*-value | 0.125 |
| alpha | 0.05 |

As the computed *p*-value is greater than the significance level alpha = 0.05, the null hypothesis H0 cannot be rejected. The risk of the null hypothesis H0 being rejected when it is true is 12.50%. If 7 instead of 10 tests are considered, the results are different, as follows:

| | |
|---|---|
| V. | 15 |
| V(standard variance) | 2.023 |
| Expected value | 7.500 |
| Variance(V) | 13.750 |
| *p*-value | 0.043 |
| alpha | 0.05 |

As the computed *p*-value is lower than the *p*-value with the significance level alpha = 0.10, the null hypothesis H0 should be rejected, and the alternative hypothesis H1, accepted. The risk the null hypothesis H0 being rejected when it is true is less than 4.31 %.

Table 9 summarizes the *p*-value variations, depending on number of samples, with a fixed significance level.

**Table 9.** *p*-value for Mann–Whitney test with a significance level of 0.05 depending on number of tests.

| *p*-Value | Number of Tests | H0 Should be Rejected | Significance |
|---|---|---|---|
| 0.012 | 10 | YES | 0.05 |
| 0.043 | 7 | YES | 0.05 |
| | 5 | NO | 0.05 |

Finally, the statistical result obtained was validated by means of the Mann–Whitney U-Test and Wilcoxon Rank Sum Test, the results of which are shown in this section and in Table 9. These studies make it possible to check the frequency distributions by comparing groups A and B, and in short, to check whether the results obtained would be the same by carrying out another number of tests. According to the results shown in Table 9, we can be conclude that the *p*-value increases as the number of tests drops, and at least 7 tests are required for the H0 to be rejected.

In the specific environment in which the GESTAC_APP was applied, the GESTAC model improved the incident resolution time by recording the logged incidents and their relationships to the captured knowledge. The validation only accounted for use cases in a single IT domain. This is only a preliminary result since the validation is incomplete. Future lines of research will apply the model in other problem domains to continue its validation. It is also planned to apply GESTAC to other environments and groups to validate the GESTAC_APP tool. According to the conducted validations, the GESTAC model already achieved impressive results for the specific domain and use case that was analyzed.

### 6.7. Qualitative Validations

To round out the abovementioned quantitative validations, the technicians were asked to evaluate incident resolution with and without the use of GESTAC. The results are shown in Table 10. The possible ratings ranged from 1 to 10, where 10 was the best score.

**Table 10.** Qualitative rating of incident resolution time with and without GESTAC.

| Expert | Evaluation Using GESTAC Prototype (from 1 to 10) | Evaluation without the GESTAC Prototype (from 1 to 10) |
|---|---|---|
| 1 | 6 | 4 |
| 2 | 8 | 4 |
| 3 | 5 | 5 |
| 4 | 6 | 6 |
| 5 | 9 | 7 |
| 6 | 8 | 4 |
| 7 | 8 | 6 |
| 8 | 7 | 5 |
| 9 | 9 | 5 |
| 10 | 6 | 4 |

*6.8. Discussion and Limitations of the Validation*

The GESTAC validation focused on the resolution of 10 incidents logged by users in different IT areas within a business domain by two different groups of technicians. One group (Technicians A) did not use GESTAC, whereas the other group applied GESTAC (Technicians B) (see Figure 1). Briefly the results were as follows:

A. The incident resolution times for Group B were shorter than for Group A.

B. The mean time was greater for Group A than for Group B, which means that the center of the distribution will be greater for Group A than for Group B.

C. The standard deviation for Group A (43) is greater than for Group B (23.026). Therefore, the data for Group A are less reliable than for Group B as their dispersion with respect to the mean covers a wider range of values.

D. There is evidence to reject H0 as the statistical value is less than the possible t-value at a significance level of 0.10. There is no evidence to reject H0 at the 0.05 significance level (Figure 17).

E. Applying the Mann–Whitney test, the *p*-value increases as the number of cases decreases. This means that, in this case, up to 7 cases have to be taken into account to gather evidence to reject H0 (Table 9).

This is the extent of the results of the validation until the model can be used in more real business situations.

**7. Conclusions**

The GESTAC model applies each of the tacit knowledge management phases. It achieves the objective of searching for solutions for business incidents logged by users, speeding up their resolution times. It identifies the sources that could provide the solutions and captures their tacit knowledge. It stores this knowledge in a KB and distributes it in the business domain for use by all authorized users. The knowledge applied as solutions to incidents is permanently evaluated and can be replaced if better sources or knowledge emerge. A prototype was implemented with a front-end and back-end architecture using software agents that were based on data input by users. The software agents can locate new and better sources and knowledge by executing deliberative processes to output results without human intervention.

The communication between human agents and the system is carried out by means of software agents, including Java coding and JSON strings. This facilitates their application in distributed and changing environments.

The choice of items for evaluating the sources and knowledge emerged from the combination of results yielded by surveys taken by knowledge managers from different companies and from the results tested on other confidence models.

The experiments conducted on a prototype using tacit knowledge management implemented as a multiagent system demonstrated that the model's biggest advantage was that it led to shorter incident resolution times for the technicians that used the prototype compared with technicians that did not use the application for the tested domain and case. In addition to the reduction of incident resolution times,

the system has another important advantage: the global count of benefits versus costs. The cost/hour of the technicians of small and medium enterprises (adding dedication, training of use in GESTAC, etc.) added to the cost of the development and implementation, deployment and execution of GESTAC is clearly lower than the costs of technicians in the event of not using this model. The calculations show relevant net benefit values from the first quarter of implementation, which is undoubtedly a second advantage to be taken into account. In a nutshell, these solutions can be improved using software agents to automatically search, evaluate and access databases based on the original data and fine-tune processes if better sources and knowledge emerge. Software agents permanently interact with each other and with human beings, transferring information. They are each experts in their role.

By capturing and conserving the tacit knowledge of the company employees, even after they leave, GESTAC aims to contribute to the development of the organizational memory and user participation by measuring their satisfaction with the solutions offered and indirectly with the sources that provided those solutions. This should act as an incentive for the improvement of the work performed in the business domain, we consider that the research achieved the stated goals, and is worth continuing to achieve further developments and improvement in the future. It would be very beneficial to also apply the concept of imprecision used to record incident severity to the knowledge ratings. In this manner, it would be possible to apply the same piece of knowledge to other similar problems.

The study carried out makes it possible to check the applicability and viability of the model presented, and its suitability for the problem to be tackled, but it has some limitations, already introduced succinctly in the previous section. Once the prototype has been validated, its use should be implemented in more varied and transversal work areas, as well as increasing the number of system test cases and thus achieving a more complete validation of the system. The sample of technicians should be increased in later studies, as well as their roles and areas of experience. This will make it possible to extrapolate the lessons learned to other business areas where tacit knowledge may represent usable intellectual capital in the company. These subsequent studies may make it necessary to adjust the overall design of the system to fit the particularities of other domains.

## 8. Future Trends

Another issue worth addressing is the generation of a new version of the interface paying more attention to visual design. In this prototype, this was a secondary issue, as the focus was on process representation. The prototype is capable of generating production-oriented software targeting security, robustness and performance. A future extension of the database agents could improve performance, as being a prototype, no load testing was conducted. Modifications could be made with a view to deploying the application on a proper web server, substituting the embedded Spring Boot solution. This could lead to possible query optimizations, as well as the use of remote agents to distribute current agents at different locations. Data-mining tools, like Business Intelligence, could then be applied at later stages of the research for the purpose of forecasting and improving the model. In short, both the model and the implementation of the GESTAC prototype were designed for large-scale extension in future versions.

Fuzzy logic could be employed to search for user problems, particularly with respect to severity. The fuzzy concept is concerned with the definition of variable degrees of any attribute [25]. These concepts could also be used to add problem-solving knowledge to determine whether it is potentially applicable to other similar problems. Problem entry should not be limited to a single area, subject, incident and severity. The current domain structure supports the possibility of several areas, subjects and incidents per problem, but the interface and primarily the system logic should be extended. This point is related to the use of fuzzy logic. It should be possible to associate a system source with more than one area in order to better represent the reality of some organizations (some sources play the role of expert in more than one area).

It is important to further explore the use of the Jade agent container and the possibility of using remote containers to distribute the agents in different locations. Additionally, this research as part of

this project has suggested that the text messages used by agents to communicate should be used to send messages to other systems which could be sent messages and use their services. The model is designed for use in other problem domains. But the GESTAC model has already achieved impressive results in for the specific use case in this particular domain, which warrant publication.

In any case all these improvements, at the interface level, databases, system back-end, fuzzy logic in the reasoning of the model, etc., can improve some of its features, but can also have a negative impact on the benefit/cost ratio, so this aspect will be considered when choosing which improvements to implement in the future.

When the model is published and disseminated for widespread use in the productive sector, a more thorough and complete test bench will be developed.

The next natural step in research, in addition to the previous ones, is also to apply this model to other problem domains, test it in them and be able to adapt it to generalize its applicability. The first concept tests have been carried out, applying "design thinking" methodologies to apply this model to other related areas such as operation support systems (OSS) for the resolution of technical incidents in ISP (Internet service providers) companies, B2B CRM, B2C and at the level of internal management and ITIL-type management in information and communication technology (ICT) companies.

**Author Contributions:** Data curation, L.P. and D.L.; Investigation, D.L.; Methodology, D.L; Supervision, G.L.; Writing—original draft, L.P. and J.L.; Writing—review & editing, D.L. and J.L.

**Funding:** This research received no external funding.

**Acknowledgments:** This work has been developed under the project called "Creation of a characterization framework for the quantification of quality in the new paradigms of web development", with identifier FH 2017–055. We would like to thank Fernando Alonso Amo from School of Computer Engineering, Universidad Politécnica de Madrid, for his constant and valuable support throughout the research, Federico Barbieri, from the School of Engineering, Universidad ORT Uruguay, Program Analysts Mauricio and Alejandro Vivanco from the School of Technology, Universidad ORT Uruguay, Lucia Vivanco, from School of Sciences UDELAR and everyone who participated in the validation and experimentation: IT and SIPI personnel at INAU, Universidad ORT Uruguay, ANCAP, DGI, the President of the Republic's Office, Seguro Americano.

**Conflicts of Interest:** The authors declare no conflict of interest.

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
