# Peer review of "A Multiagent System Prototype of a Tacit Knowledge Management Model to Reduce Labor Incident Resolution Times"

_applsci, doi:10.3390/app9245448_

Round 1
Reviewer 1 Report
A) Broad comments
A1. Definition of tacit knowledge
In literature the concept of tacit knowledge has been defined and used in many ways; since it belongs to the main topic of this paper it would be useful to have a dedicated section in which the authors describe their view of this concept, not simply listing various views that they in some way consider acceptable but putting together the parts to build a whole.
A2. Model description
In the first part of the paper the GESTAC model is described among others by a table showing "the differences between each of the models and the GESTAC model". (171)
Up to this point the reader has seen only claims ABOUT the model (claims about features, strenghts, differences from other models, etc.) but the model itself is still unknown to the reader, is an X. So, all the claims are claims about X.
Consider changing the order:
a) first describe the model itself (X = WHAT) => after 221
b) then present your claims.
In this way your claims can (and should) be connected with the WHAT of the model: ==> Because of the WHAT "W" we can make the CLAIM "C".
A3. GESTAC Prototype
(427) In section 5 "GESTAC Prototype" you present examples of algorithms and their encodings, using about 100 lines from 487 to 590. Why are the encoding so interesting? It would be more interesting to see an example (from your experimentation) with real data (messages, etc.) and what the agents do with them.
A4. Evaluation and further developments
In the conclusion you claim that "the model’s biggest advantage was that it led to shorter incident resolution times for the technicians" (961). This is a veryattractive benefit, but what about "costs" and the benefit/cost ration? Could you elaborate on this aspect?
In section 8 (975) you mention some interesting improvements to be addressed in future work, like user interface, database agents, proper web server design, data mining, fuzzy logic, etc. This work would improve the benefits but the danger is, that it could reduce the benefit/cost ratio, so this aspect should be considered when selecting which improvements to address first.
B) Specific comments
(32) "implemented mechanically": what do you mean? please explain.
(42,43) "tacit knowledge management model was designed, called GESTAC": please explain what mean the letters in the acronym GESTAC. Probably GES => from spanish "GEStion" and TAC => from spanish "TACita"; but should be explicitly described, not tacitly ;-)
(413) " a) by personally to broach the issue and, b) by email ...": what means this sentence? I do not understand this grammar (sequence of preposition, adverb, infinitive verb, article, object.) nor this semantics ("by personally" and "by email" are two very different semantic levels).
You mean "a) by personally pushing forward the issue ... b) by answering questions of an email survey ..." ?
Author Response
In the following we have listed all the changes and modifications made during the revision of the article based on the reviewers' comments. We are immensely grateful to the reviewers for their effort and commitment, which have unquestionably helped to notably improve the original paper. Below we reproduce their comments word for word, specifying the changes made to remedy the problem raised underneath.
Comment 1:
A1. Definition of tacit knowledge
In literature the concept of tacit knowledge has been defined and used in many ways; since it belongs to the main topic of this paper it would be useful to have a dedicated section in which the authors describe their view of this concept, not simply listing various views that they in some way consider acceptable but putting together the parts to build a whole.
Response 1:
We fully agree with the reviewer. To improve this aspect, a small subsection has been created, numbered 2.1 Definition of tacit knowledge, which succinctly specifies the point of view of the authors on this concept so relevant in the article.
This section, included on page 3 of 39, is this one:
“2.1. Definition of tacit knowledge
The present research considers that tacit knowledge is that implicit, not formalized and not communicated wisdom that knowledge workers treasure thanks to their day-to-day experience [1]. Thanks to this knowledge, they are able to carry out tasks quickly and efficiently, without having to invest too much time in thinking about what steps to take, or how to carry them out, since they do so almost systematically.”
Comment 2:
A2. Model description
In the first part of the paper the GESTAC model is described among others by a table showing "the differences between each of the models and the GESTAC model". (171)
Up to this point the reader has seen only claims ABOUT the model (claims about features, strenghts, differences from other models, etc.) but the model itself is still unknown to the reader, is an X. So, all the claims are claims about X.
Consider changing the order:
a) first describe the model itself (X = WHAT) => after 221 b) then present your claims.In this way your claims can (and should) be connected with the WHAT of the model: ==> Because of the WHAT "W" we can make the CLAIM "C".
Response 2:
We agree with the reviewer, and indeed, using the order he proposes, the information on the model itself and its comparison with other existing proposals becomes clearer. Therefore, the entire comparative text (from line 165 to line 181) has been removed from the section "2.2 Models related to GESTAC" and moved to a new sub-section, “3.9 Comparative of existing knowledge management models”. This section, located in line 369, is much clearer, as this information is better contextualized because the reader already knows which model the article proposes and knows its characteristics.
Comment 3:
A3. GESTAC Prototype
(427) In section 5 "GESTAC Prototype" you present examples of algorithms and their encodings, using about 100 lines from 487 to 590. Why are the encoding so interesting? It would be more interesting to see an example (from your experimentation) with real data (messages, etc.) and what the agents do with them.
Response 3:
We agree with this comment, which undoubtedly leads to an improvement in the quality of the article. In the light of the other reviews received, other experts do seem to consider the content described in the algorithms to be relevant. Likewise, they consider the article to be "a little long", with which it has been necessary to make a compromise decision, and to include real examples of messages, executions and agent interactions without disregarding the algorithms or overextending the article. For this purpose, all this example material, extracted directly from the experimentation carried out, has been included in the research project's web site, reference [37]. The article refers to this new content right at the end of section 5.3 (after figure 11) where this information has been included:
“Once these three algorithms have been presented, it is possible to obtain information on examples of behaviour, data, messages and real system executions, through the content available in [37].”
Comment 4:
A4. Evaluation and further developments
In the conclusion you claim that "the model’s biggest advantage was that it led to shorter incident resolution times for the technicians" (961). This is a very attractive benefit, but what about "costs" and the benefit/cost ration? Could you elaborate on this aspect?
Response 4:
Indeed, we agree with the reviewer's assessment, and beyond minimizing incident resolution times, a clear advantage arises from using the system: costs in the company, both for the reduction of resolution times and for the hourly costs of the technicians involved, were significantly reduced in the SMEs where the experimentation was carried out. For this reason, information on this important aspect (omitted in the previous version) has been included in the fourth paragraph of the conclusions (line 990), just after the statement indicated in the commentary. The added content is this:
"[...]. In addition to the reduction of incident resolution times, the system has another important advantage: the global count of benefits versus costs. The cost/hour of the technicians of small and medium enterprises (adding dedication, training of use in GESTAC, etc.) added to the cost of the development and implementation, deployment and execution of GESTAC is clearly lower than the costs of technicians in the event of not using this model. The calculations show relevant net benefit values from the first quarter of implementation, which is undoubtedly a second advantage to be taken into account.[...]".
Comment 5:
In section 8 (975) you mention some interesting improvements to be addressed in future work, like user interface, database agents, proper web server design, data mining, fuzzy logic, etc. This work would improve the benefits but the danger is, that it could reduce the benefit/cost ratio, so this aspect should be considered when selecting which improvements to address first.
Response 5:
Following on from the previous comment, this aspect also seems relevant to us, and we had not included it in the first version of the article. We now mention this in section 8, at the end of the third paragraph, line 1039. In that text we have added this in order to clarify it:
"In any case all these improvements, at the interface level, databases, system back-end, fuzzy logic in the reasoning of the model, etc., can improve some of its features, but can also have a negative impact on the benefit/cost ratio, so this aspect will be considered when choosing which improvements to implement in the future.”
Comment 6:
(32) "implemented mechanically": what do you mean? please explain.
Response 6:
It was an errata in the translation, it has been changed to:
"applied automatically (almost unconsciously)".
Comment 7:
(42,43) "tacit knowledge management model was designed, called GESTAC": please explain what mean the letters in the acronym GESTAC. Probably GES => from spanish "GEStion" and TAC => from spanish "TACita"; but should be explicitly described, not tacitly ;-)
Response 7:
The reviser is right, in its origin, the acronym used to name the model comes from the union of the first two syllables of two Spanish words: gestión (management) and tácito ( tacit).
This fact is now clarified "explicitly" in the text itself, in line 45 thanks to this text:
"name that originally resulted from joining the first two syllables of two Spanish words, “gestión” (management) and “tácito” (tacit), which succinctly define the objective of the model".
Comment 8:
(413) " a) by personally to broach the issue and, b) by email ...": what means this sentence? I do not understand this grammar (sequence of preposition, adverb, infinitive verb, article, object.) nor this semantics ("by personally" and "by email" are two very different semantic levels).
You mean "a) by personally pushing forward the issue ... b) by answering questions of an email survey ..." ?
Response 8:
We fully agree with the reviewer, and yes, the phrases he himself proposes are what we referred to in our text. Both sentences have now been replaced, in line 417, by:
"a) by personally pushing forward the issue and, b) by answering questions of an email survey regarding rating sources of knowledge”.
Reviewer 2 Report
This is an interesting piece of work, applying an agent-based approach to the problem of using tacit knowledge, by identifying the people in an organisation who are sources of knowledge, and storing the used knowledge in the resolution of incidents.
The paper focuses mostly on the architecture and on evaluation of a test conducted in an organisation, where technicians were asked to solve some incidents in different areas, of their competence, with or without recurring to the system.
The results seem to support the conclusion that the system can be effective in concrete situation.
In general terms is well written and well-organised. A few points could improve the paper.
Section 5.2 discusses the rationale for using agents, but the kind of argument there could apply to any form of modular organisation of the system, so some more compelling reason should be given.
In the class diagram of Figure 15, why is there a multiplicity of 1 at the Knowledge end of the association with Source? Could we not have that a Source is a source for several different pieces of knowledge?
In general, it would be nice to see a concrete example, even taken from the scenarios used in the tests, and show how Knowledge, Source, Area, etc. would be instantiatied in this case, also showing the results of applying the selection algorithms on the adopted scenario.
Author Response
In the following we have listed all the changes and modifications made during the revision of the article based on the reviewers' comments. We are immensely grateful to the reviewers for their effort and commitment, which have unquestionably helped to notably improve the original paper. Below we reproduce their comments word for word, specifying the changes made to remedy the problem raised underneath.
Comment:
This is an interesting piece of work, applying an agent-based approach to the problem of using tacit knowledge, by identifying the people in an organisation who are sources of knowledge, and storing the used knowledge in the resolution of incidents.
The paper focuses mostly on the architecture and on evaluation of a test conducted in an organisation, where technicians were asked to solve some incidents in different areas, of their competence, with or without recurring to the system.
The results seem to support the conclusion that the system can be effective in concrete situation.
In general terms is well written and well-organised. A few points could improve the paper.
Comment 1:
Section 5.2 discusses the rationale for using agents, but the kind of argument there could apply to any form of modular organisation of the system, so some more compelling reason should be given.
Response 1:
Indeed the explanation included in point 5.2 showed some achievable advantages when using agents, but which could be extrapolated when using any modular organisational solution for the system. A new paragraph has been included justifying the choice of the agent-based programming paradigm, based on the proof of concept carried out during the first months of project development. The paragraph included, just before heading 5.3, is as follows:
"In general, other forms of modular organization of the system could have been used instead of using agents, but in the early stages of the project it could be verified that the prototypes developed with agents were ahead of the use of other technologies: they offered better results in terms of maintainability, computational efficiency and adaptability in contexts of changing information like this one. Thus, the intervention of expert domain users was minimized, thanks to the flexible, reactive, proactive and social behavior of the agents".
Comment 2:
In the class diagram of Figure 15, why is there a multiplicity of 1 at the Knowledge end of the association with Source? Could we not have that a Source is a source for several different pieces of knowledge?
Response 2:
That is right, we can have a Source that originates several different pieces of Knowledge, something in fact habitual in case the source of this information is sufficiently broad and/or transversal. It was therefore an error in the class diagram that we have corrected (see new figure 15 of the paper).
Comment 3:
In general, it would be nice to see a concrete example, even taken from the scenarios used in the tests, and show how Knowledge, Source, Area, etc. would be instantiatied in this case, also showing the results of applying the selection algorithms on the adopted scenario.
Response 3:
We agree with the reviewer, and we have created and updated web content with concrete information of the execution of the example perfomed in the experimentation carried out for the article, accessible in the site referenced in reference 37, fact that we have indicated in the corresponding section (at the end of section 5.6):
“It is possible to consult concrete examples of instances of the class diagram present in Figure 15 in real scenarios such as that described in section 6 through the on-line repository [37].”
The publication of some instances of knowledge, area, source, etc. are subject to the consent and proper processing of personal data that we have already requested from the applicant companies. As soon as they give us the necessary approval, we will update and include more information available on this site.
Reviewer 3 Report
The introduction should be changed. Now it is starting from the description of the section but is should first introduce the topic of the paper and the next problem statement and in the end, it should be described the structure of the paper.
In conclusion, the limitations of the study should be presented.
Author Response
Comment 1:
The introduction should be changed. Now it is starting from the description of the section but is should first introduce the topic of the paper and the next problem statement and in the end, it should be described the structure of the paper.
Response 1:
We fully agree with the reviewer, and we thank him very much for his comments, which have undoubtedly made it possible to improve the article. We have redrafted the introduction, changing the first paragraph of this section and the last. Now the Introduction begins with the problem addressed by the research and the topic addressed by the article, including this paragraph:
" The tacit knowledge of the human resources of companies is a precious asset for them, which often cannot be exploited in an adequate time and manner, because it is implicit, not formalized and not communicated. This article proposes a model to transform this tacit knowledge into permanent organizational capital, properly maintained and available, and validates this model in an area where this knowledge is especially necessary: the resolution of labor incidents. The research will focus on […]".
And this section ends with a summary of the general structure of the article, with this paragraph:
"The paper is structured in seven sections. Section 2 describes the theoretical groundwork underlying the research and some of the knowledge management models used in business domains. It highlights the differences between previous models and the proposed GESTAC model. Section 3 describes materials, methods, the GESTAC model, its objective, the hypothesis to be tested, its design features, components and how they interact, as well as the processes and associated calculations for evaluating and continually reevaluating knowledge and knowledge sources. Section 4 details the methodology used throughout the incident handling processes, and the model design, layers, objectives and activities. Section 5 explains GESTAC Model Prototype, its objective , its architecture, the reason for using the multiagent systems paradigm, the software agents used, the logic of some of the algorithms and their main encodings. It also provides specific examples of inter-agent communication, and addresses agent interaction, domain layer agents, the data access agent, and the class diagram. Section 6 specifies the aims of the experimentation, its theoretical background, its design, the incidents used in the experiment and the analysis of the results according to different statistical tests to accept or reject the proposed hypothesis. Section 7 analyze results, whether or not the implemented prototype meets the original objective and the project generated advantages in the business world. Section 8 describes possible future lines of research concerning the GESTAC model".
Comment 2:
In conclusion, the limitations of the study should be presented.
Response 2:
We agree with the reviewer, and have included a paragraph discussing the limitations of the study conducted just at the end of section 7, in order to delimit the scope of the results and conclusions obtained. This paragraph included is this one:
" The study carried out makes it possible to check the applicability and viability of the model presented, and its suitability for the problem to be tackled, but it has some limitations, already introduced succinctly in the previous section. Once the prototype has been validated, its use should be implemented in more varied and transversal work areas, as well as increasing the number of system test cases and thus achieving a more complete validation of the system. The sample of technicians should be increased in later studies, as well as their roles and areas of experience. This will make it possible to extrapolate the lessons learned to other business areas where tacit knowledge may represent usable intellectual capital in the company. These subsequent studies may make it necessary to adjust the overall design of the system to fit the particularities of other domains".
Reviewer 4 Report
I enjoyed reading the paper, although it is too long. Anyway, the fact that it was interesting, helped.
The paper is well structured and addresses an important topic - how to capture and manage tacit knowledge. It presents a model, created by the authors, and then tested in the IT area. Results show its usefulness.
There are no major weaknesses. Although, the presentation of the statistical part of the research could be improved (suggestion) as the reader, in some parts, has some difficulty in following and understanding the reasoning. I also wonder if the author pretends to test the model in other areas. Anyway, this is a good paper and I congratulate the authors.
Author Response
In the following we have listed all the changes and modifications made during the revision of the article based on the reviewers' comments. We are immensely grateful to the reviewers for their effort and commitment, which have unquestionably helped to notably improve the original paper. Below we reproduce their comments word for word, specifying the changes made to remedy the problem raised underneath.
Comment:
I enjoyed reading the paper, although it is too long. Anyway, the fact that it was interesting, helped.
The paper is well structured and addresses an important topic - how to capture and manage tacit knowledge. It presents a model, created by the authors, and then tested in the IT area. Results show its usefulness.
Comment 1:
There are no major weaknesses. Although, the presentation of the statistical part of the research could be improved (suggestion) as the reader, in some parts, has some difficulty in following and understanding the reasoning.
Response 1:
We agree with the reviewer, and throughout section 6.5 we have added small explanatory paragraphs about the reason, object, results and reading of the different statistical studies (tables from 3 to 5 and from 8 to 9) so that in each one of them it is a question of guiding the reader a little more to facilitate its understanding.
The added paragraphs are:
- Table 3: " For the calculation of resolution time, mean and variance, the basic statistics have simply been obtained from the data collected in table 2 for both groups (A and B).", page 25 of 39.
- Table 4: " Thanks to the use of the student T, it is intended to check whether the average difference in the resolution times obtained in table 3 is statistically significant or not. For this purpose, a study was carried out with a significance level of 0.05 and 0.1, shown in table 4.", page 25 of 39.
- Table 5: " Table 5 illustrates the decision values for the study carried out, in such a way that in the light of the data obtained the initial hypothesis must be rejected (groups A and B had equivalent results) and the hypothesis that the users who used GESTAC spent significantly less time must be accepted.", page 26 of 39.
- Table 8: " Table 8 includes a statistical study for the sign test. This study is carried out to validate the experiment, and to verify that the incidents managed by both groups were equivalent in typology and type of resolution.", page 28 of 39 .
- Table 9: " Finally, the statistical result obtained was validated by means of two tests called Mann-Whitney U test and Wilcoxon Rank Sum Test, the results of which are shown in this section and in table 9. These studies make it possible to check the frequency distributions by comparing groups A and B, and in short, to check whether the results obtained would be the same by carrying out another number of tests. ", pag 31 of 39.
Comment 2:
I also wonder if the author pretends to test the model in other areas. Anyway, this is a good paper and I congratulate the authors.
Response 2:
Effectively, our objective, once the model presented in this article is made public, is to apply it to other related areas where it may be useful. To clarify, we have added the following paragraph in section 8, just before the acknowledgements section:
" The next natural step in research, in addition to the previous ones, is also to apply this model to other problem domains, test it in them and be able to adapt it to generalize its applicability. The first concept tests have been carried out, applying "design thinking" methodologies to apply this model to other related areas such as operation support systems (OSS) for the resolution of technical incidents in ISP companies (Internet Service Providers), B2B CRM, B2C and at the level of internal management and ITIL type management in ICT companies".